# There are Many Consistent Explanations of Unlabeled Data: Why You Should Average

**Ben Athiwaratkun, Marc Finzi, Pavel Izmailov, Andrew Gordon Wilson**
`{pa338, maf388, pi49, andrew}@cornell.edu`
Cornell University

## Abstract

Presently the most successful approaches to semi-supervised learning are based on *consistency regularization*, whereby a model is trained to be robust to small perturbations of its inputs and parameters. To understand consistency regularization, we conceptually explore how loss geometry interacts with training procedures. The consistency loss dramatically improves generalization performance over supervised-only training; however, we show that SGD struggles to converge on the consistency loss and continues to make large steps that lead to changes in predictions on the test data. Motivated by these observations, we propose to train consistency-based methods with Stochastic Weight Averaging (SWA), a recent approach which averages weights along the trajectory of SGD with a modified learning rate schedule. We also propose *fast-SWA*, which further accelerates convergence by averaging multiple points within each cycle of a cyclical learning rate schedule. With weight averaging, we achieve the best known semi-supervised results on CIFAR-10 and CIFAR-100, over many different quantities of labeled training data. For example, we achieve 5.0% error on CIFAR-10 with only 4000 labels, compared to the previous best result in the literature of 6.3%.

## 1 Introduction

Recent advances in deep unsupervised learning, such as generative adversarial networks (GANs) (Goodfellow et al., 2014), have led to an explosion of interest in semi-supervised learning. Semi-supervised methods make use of both unlabeled and labeled training data to improve performance over purely supervised methods. Semi-supervised learning is particularly valuable in applications such as medical imaging, where labeled data may be scarce and expensive (Oliver et al., 2018).

Currently the best semi-supervised results are obtained by *consistency-enforcing* approaches (Bachman et al., 2014; Laine and Aila, 2017; Tarvainen and Valpola, 2017; Miyato et al., 2017; Park et al., 2017). These methods use unlabeled data to stabilize their predictions under input or weight perturbations. Consistency-enforcing methods can be used at scale with state-of-the-art architectures. For example, the recent Mean Teacher (Tarvainen and Valpola, 2017) model has been used with the Shake-Shake (Gastaldi, 2017) architecture and has achieved the best semi-supervised performance on the consequential CIFAR benchmarks.

This paper is about conceptually understanding and improving consistency-based semi-supervised learning methods. Our approach can be used as a guide for exploring how loss geometry interacts with training procedures in general. We provide several novel observations about the training objective and optimization trajectories of the popular $\Pi$ (Laine and Aila, 2017) and Mean Teacher (Tarvainen and Valpola, 2017) consistency-based models. Inspired by these findings, we propose to improve SGD solutions via stochastic weight averaging (SWA) (Izmailov et al., 2018), a recent method that averages weights of the networks corresponding to different training epochs to obtain a single model with improved generalization. On a thorough empirical study we show that this procedure achieves the best known semi-supervised results on consequential benchmarks. In particular:

- We show in Section 3.1 that a simplified $\Pi$ model implicitly regularizes the norm of the Jacobian of the network outputs with respect to both its inputs and its weights, which in turn encourages flatter solutions. Both the reduced Jacobian norm and flatness of solutions have been related to generalization in the literature (Sokolić et al., 2017; Novak et al., 2018; Chaudhari et al.,

2016; Schmidhuber and Hochreiter, 1997; Keskar et al., 2017; Izmailov et al., 2018). Interpolating between the weights corresponding to different epochs of training we demonstrate that the solutions of $\Pi$ and Mean Teacher models are indeed flatter along these directions (Figure 1b).

- In Section 3.2, we compare the training trajectories of the $\Pi$, Mean Teacher, and supervised models and find that the distances between the weights corresponding to different epochs are much larger for the consistency based models. The error curves of consistency models are also wider (Figure 1b), which can be explained by the flatness of the solutions discussed in section 3.1. Further we observe that the predictions of the SGD iterates can differ significantly between different iterations of SGD.

- We observe that for consistency-based methods, SGD does not converge to a single point but continues to explore many solutions with high distances apart. Inspired by this observation, we propose to average the weights corresponding to SGD iterates, or ensemble the predictions of the models corresponding to these weights. Averaging weights of SGD iterates compensates for larger steps, stabilizes SGD trajectories and obtains a solution that is centered in a flat region of the loss (as a function of weights). Further, we show that the SGD iterates correspond to models with diverse predictions – using weight averaging or ensembling allows us to make use of the improved diversity and obtain a better solution compared to the SGD iterates. In Section 3.3 we demonstrate that both ensembling predictions and averaging weights of the networks corresponding to different training epochs significantly improve generalization performance and find that the improvement is much larger for the $\Pi$ and Mean Teacher models compared to supervised training. We find that averaging weights provides similar or improved accuracy compared to ensembling, while offering the computational benefits and convenience of working with a single model. Thus, we focus on weight averaging for the remainder of the paper.

- Motivated by our observations in Section 3 we propose to apply Stochastic Weight Averaging (SWA) (Izmailov et al., 2018) to the $\Pi$ and Mean Teacher models. Based on our results in Section 3.3 we propose several modifications to SWA in Section 4. In particular, we propose fast-SWA, which (1) uses a learning rate schedule with longer cycles to increase the distance between the weights that are averaged and the diversity of the corresponding predictions; and (2) averages weights of multiple networks within each cycle (while SWA only averages weights corresponding to the lowest values of the learning rate within each cycle). In Section 5, we show that fast-SWA converges to a good solution much faster than SWA.

- Applying weight averaging to the $\Pi$ and Mean Teacher models we improve the best reported results on CIFAR-10 for $1k$, $2k$, $4k$ and $10k$ labeled examples, as well as on CIFAR-100 with $10k$ labeled examples. For example, we obtain $5.0\%$ error on CIFAR-10 with only $4k$ labels, improving the best result reported in the literature (Tarvainen and Valpola, 2017) by $1.3\%$. We also apply weight averaging to a state-of-the-art domain adaptation technique (French et al., 2018) closely related to the Mean Teacher model and improve the best reported results on domain adaptation from CIFAR-10 to STL from $19.9\%$ to $16.8\%$ error.

- We release our code at https://github.com/benathi/fastswa-semi-sup

## 2 BACKGROUND

### 2.1 CONSISTENCY BASED MODELS

We briefly review semi-supervised learning with consistency-based models. This class of models encourages predictions to stay similar under small perturbations of inputs or network parameters. For instance, two different translations of the same image should result in similar predicted probabilities. The consistency of a model (student) can be measured against its own predictions (e.g. $\Pi$ model) or predictions of a different teacher network (e.g. Mean Teacher model). In both cases we will say a *student* network measures consistency against a *teacher* network.

**Consistency Loss** In the semi-supervised setting, we have access to labeled data $\mathcal{D}_L = \{(x_i^L, y_i^L)\}_{i=1}^{N_L}$, and unlabeled data $\mathcal{D}_U = \{x_i^U\}_{i=1}^{N_U}$.

Given two perturbed inputs $x'$, $x''$ of $x$ and the perturbed weights $w_f'$ and $w_g'$, the consistency loss penalizes the difference between the *student*'s predicted probablities $f(x'; w_f')$ and the *teacher*'s

$g(x''; w'_g)$. This loss is typically the Mean Squared Error or KL divergence:

$$\ell_{\text{cons}}^{\text{MSE}}(w_f, x) = \|f(x'; w'_f) - g(x'', w'_g)\|^2 \text{ or } \ell_{\text{cons}}^{\text{KL}}(w_f, x) = \text{KL}(f(x'; w'_f)\|g(x'', w'_g)). \quad (1)$$

The total loss used to train the model can be written as

$$L(w_f) = \underbrace{\sum_{(x,y)\in\mathcal{D}_L} \ell_{\text{CE}}(w_f, x, y)}_{L_{\text{CE}}} + \lambda \underbrace{\sum_{x\in\mathcal{D}_L\cup\mathcal{D}_U} \ell_{\text{cons}}(w_f, x)}_{L_{\text{cons}}}, \quad (2)$$

where for classification $L_{\text{CE}}$ is the cross entropy between the model predictions and supervised training labels. The parameter $\lambda > 0$ controls the relative importance of the consistency term in the overall loss.

**Π Model** The Π model, introduced in Laine and Aila (2017) and Sajjadi et al. (2016), uses the student model $f$ as its own teacher. The data (input) perturbations include random translations, crops, flips and additive Gaussian noise. Binary dropout (Srivastava et al., 2014) is used for weight perturbation.

**Mean Teacher Model** The Mean Teacher model (MT) proposed in Tarvainen and Valpola (2017) uses the same data and weight perturbations as the Π model; however, the teacher weights $w^g$ are the exponential moving average (EMA) of the student weights $w^f$: $w_g^k = \alpha \cdot w_g^{k-1} + (1-\alpha) \cdot w_f^k$. The decay rate $\alpha$ is usually set between 0.9 and 0.999. The Mean Teacher model has the best known results on the CIFAR-10 semi-supervised learning benchmark (Tarvainen and Valpola, 2017).

**Other Consistency-Based Models** Temporal Ensembling (TE) (Laine and Aila, 2017) uses an exponential moving average of the student outputs as the teacher outputs in the consistency term for training. Another approach, Virtual Adversarial Training (VAT) (Miyato et al., 2017), enforces the consistency between predictions on the original data inputs and the data perturbed in an adversarial direction $x' = x + \epsilon r_{\text{adv}}$, where $r_{\text{adv}} = \arg\max_{r:\|r\|=1} \text{KL}[f(x, w)\|f(x + \xi r, w)]$.

## 3 UNDERSTANDING CONSISTENCY-ENFORCING MODELS

In Section 3.1, we study a simplified version of the Π model theoretically and show that it penalizes the norm of the Jacobian of the outputs with respect to inputs, as well as the eigenvalues of the Hessian, both of which have been related to generalization (Sokolić et al., 2017; Novak et al., 2018; Dinh et al., 2017a; Chaudhari et al., 2016). In Section 3.2 we empirically study the training trajectories of the Π and MT models and compare them to the training trajectories in supervised learning. We show that even late in training consistency-based methods make large training steps, leading to significant changes in predictions on test. In Section 3.3 we show that averaging weights or ensembling predictions of the models proposed by SGD at different training epochs can lead to substantial gains in accuracy and that these gains are much larger for Π and MT than for supervised training.

### 3.1 SIMPLIFIED Π MODEL PENALIZES LOCAL SHARPNESS

**Penalization of the input-output Jacobian norm.** Consider a simple version of the Π model, where we only apply small additive perturbations to the student inputs: $x' = x + \epsilon z$, $z \sim \mathcal{N}(0, I)$ with $\epsilon \ll 1$, and the teacher input is unchanged: $x'' = x$.[1] Then the consistency loss $\ell_{cons}$ (Eq. 1) becomes $\ell_{cons}(w, x, \epsilon) = \|f(w, x + \epsilon z) - f(w, x)\|^2$. Consider the estimator $\hat{Q} = \lim_{\epsilon\to 0} \frac{1}{\epsilon^2} \frac{1}{m} \sum_{i=1}^m \ell_{cons}(w, x_i, \epsilon)$. We show in Section A.5 that

$$\mathbb{E}[\hat{Q}] = \mathbb{E}_x[\|J_x\|_F^2] \quad \text{and} \quad \text{Var}[\hat{Q}] = \frac{1}{m}\left(\text{Var}[\|J_x\|_F^2] + 2\mathbb{E}[\|J_x^T J_x\|_F^2]\right),$$

where $J_x$ is the Jacobian of the network's outputs with respect to its inputs evaluated at $x$, $\|\cdot\|_F$ represents Frobenius norm, and the expectation $\mathbb{E}_x$ is taken over the distribution of labeled and

---

[1]This assumption can be relaxed to $x'' = x + \epsilon z_2$ $z_2 \sim \mathcal{N}(0, I)$ without changing the results of the analysis, since $\epsilon(z_1 - z_2) = 2\epsilon\bar{z}$ with $\bar{z} \sim \mathcal{N}(0, I)$.

unlabeled data. That is, $\hat{Q}$ is an unbiased estimator of $\mathbb{E}_x[\|J_x\|_F^2]$ with variance controlled by the minibatch size $m$. Therefore, the consistency loss implicitly penalizes $\mathbb{E}_x[\|J_x\|_F^2]$.

The quantity $\|J_x\|_F$ has been related to generalization both theoretically (Sokolić et al., 2017) and empirically (Novak et al., 2018). For linear models $f(x) = Wx$, penalizing $\|J_x\|_F$ exactly corresponds to weight decay, also known as $L_2$ regularization, since for linear models $J_x = W$, and $\|W\|_F^2 = \|\text{vec}(W)\|_2^2$. Penalizing $\mathbb{E}_x[\|J_x\|_F^2]$ is also closely related to the graph based (manifold) regularization in Zhu et al. (2003) which uses the graph Laplacian to approximate $\mathbb{E}_x[\|\nabla_{\mathcal{M}} f\|^2]$ for nonlinear models, making use of the manifold structure of unlabeled data.

Isotropic perturbations investigated in this simplified $\Pi$ model will not in general lie along the data manifold, and it would be more pertinent to enforce consistency to perturbations sampled from the space of natural images. In fact, we can interpret consistency with respect to standard data augmentations (which are used in practice) as penalizing the manifold Jacobian norm in the same manner as above. See Section A.5 for more details.

**Penalization of the Hessian's eigenvalues.** Now, instead of the input perturbation, consider the weight perturbation $w' = w + \epsilon z$. Similarly, the consistency loss is an unbiased estimator for $\mathbb{E}_x[\|J_w\|_F^2]$, where $J_w$ is the Jacobian of the network outputs with respect to the weights $w$. In Section A.6 we show that for the MSE loss, the expected trace of the Hessian of the loss $\mathbb{E}_x[\text{tr}(H)]$ can be decomposed into two terms, one of which is $\mathbb{E}_x[\|J_w\|_F^2]$. As minimizing the consistency loss of a simplified $\Pi$ model penalizes $\mathbb{E}_x[\|J_w\|_F^2]$, it also penalizes $\mathbb{E}_x[\text{tr}(H)]$. As pointed out in Dinh et al. (2017a) and Chaudhari et al. (2016), the eigenvalues of $H$ encode the local information about sharpness of the loss for a given solution $w$. Consequently, the quantity $\text{tr}(H)$ which is the sum of the Hessian eigenvalues is related to the notion of sharp and flat optima, which has recently gained attention as a proxy for generalization performance (see e.g. Schmidhuber and Hochreiter, 1997; Keskar et al., 2017; Izmailov et al., 2018). Thus, based on our analysis, the consistency loss in the simplified $\Pi$ model encourages flatter solutions.

### 3.2 Analysis of Solutions along SGD Trajectories

In the previous section we have seen that in a simplified $\Pi$ model, the consistency loss encourages lower input-output Jacobian norm and Hessian's eigenvalues, which are related to better generalization. In this section we analyze the properties of minimizing the consistency loss in a practical setting. Specifically, we explore the trajectories followed by SGD for the consistency-based models and compare them to the trajectories in supervised training.

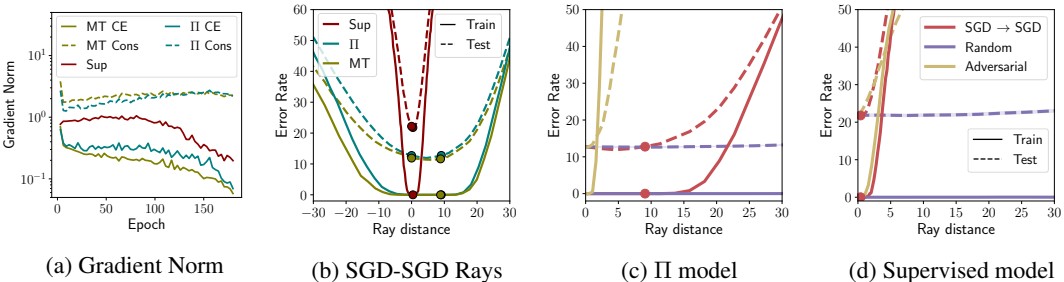

(a) Gradient Norm    (b) SGD-SGD Rays    (c) $\Pi$ model    (d) Supervised model

Figure 1: **(a)**: The evolution of the gradient norm for the consistency regularization term (Cons) and the cross-entropy term (CE) in the $\Pi$, MT, and standard supervised (CE only) models during training. **(b)**: Train and test errors along rays connecting two SGD solutions for each respective model. **(c)** and **(d)**: Comparison of errors along rays connecting two SGD solutions, random rays, and adversarial rays for the $\Pi$ and supervised models. See Section A.1 for the analogous Mean Teacher model's plot.

We train our models on CIFAR-10 using $4k$ labeled data for $180$ epochs. The $\Pi$ and Mean Teacher models use $46k$ data points as unlabeled data (see Sections A.8 and A.9 for details). First, in Figure 1a we visualize the evolution of norms of the gradients of the cross-entropy term $\|\nabla L_{\text{CE}}\|$ and consistency term $\|\nabla L_{\text{cons}}\|$ along the trajectories of the $\Pi$, MT, and standard supervised models (using CE loss only). We observe that $\|\nabla L_{\text{Cons}}\|$ remains high until the end of training and dominates the gradient $\|\nabla L_{\text{CE}}\|$ of the cross-entropy term for the $\Pi$ and MT models. Further, for both the $\Pi$

and MT models, $\|\nabla L_{\text{Cons}}\|$ is much larger than in supervised training implying that the $\Pi$ and MT models are making substantially larger steps until the end of training. These larger steps suggest that rather than converging to a single minimizer, SGD continues to actively explore a large set of solutions when applied to consistency-based methods.

For further understand this observation, we analyze the behavior of train and test errors in the region of weight space around the solutions of the $\Pi$ and Mean Teacher models. First, we consider the one-dimensional rays $\phi(t) = t \cdot w_{180} + (1-t)w_{170}$, $t \geq 0$, connecting the weight vectors $w_{170}$ and $w_{180}$ corresponding to epochs 170 and 180 of training. We visualize the train and test errors (measured on the labeled data) as functions of the distance from the weights $w_{170}$ in Figure 1b. We observe that the distance between the weight vectors $w_{170}$ and $w_{180}$ is much larger for the semi-supervised methods compared to supervised training, which is consistent with our observation that the gradient norms are larger which implies larger steps during optimization in the $\Pi$ and MT models. Further, we observe that the train and test error surfaces are much wider along the directions connecting $w_{170}$ and $w_{180}$ for the consistency-based methods compared to supervised training. One possible explanation for the increased width is the effect of the consistency loss on the Jacobian of the network and the eigenvalues of the Hessian of the loss discussed in Section 3.1. We also observe that the test errors of interpolated weights can be lower than errors of the two SGD solutions between which we interpolate. This error reduction is larger in the consistency models (Figure 1b).

We also analyze the error surfaces along *random* and *adversarial* rays starting at the SGD solution $w_{180}$ for each model. For the random rays we sample 5 random vectors $d$ from the unit sphere and calculate the average train and test errors of the network with weights $w_{t_1} + sd$ for $s \in [0, 30]$. With adversarial rays we evaluate the error along the directions of the fastest ascent of test or train loss $d_{adv} = \frac{\nabla L_{CE}}{\|\nabla L_{CE}\|}$. We observe that while the solutions of the $\Pi$ and MT models are much wider than supervised training solutions along the SGD-SGD directions (Figure 1b), their widths along random and adversarial rays are comparable (Figure 1c, 1d)

We analyze the error along SGD-SGD rays for two reasons. Firstly, in fast-SWA we are averaging solutions traversed by SGD, so the rays connecting SGD iterates serve as a proxy for the space we average over. Secondly, we are interested in evaluating the width of the solutions that we explore during training which we expect will be improved by the consistency training, as discussed in Section 3.1 and A.6. We expect width along random rays to be less meaningful because there are many directions in the parameter space that do not change the network outputs (Dinh et al., 2017b; Gur-Ari et al., 2018; Sagun et al., 2017). However, by evaluating SGD-SGD rays, we can expect that these directions corresponds to meaningful changes to our model because individual SGD updates correspond to directions that change the predictions on the training set. Furthermore, we observe that different SGD iterates produce significantly different predictions on the test data.

Neural networks in general are known to be resilient to noise, explaining why both MT, $\Pi$ and supervised models are flat along random directions (Arora et al., 2018). At the same time neural networks are susceptible to targeted perturbations (such as adversarial attacks). We hypothesize that we do not observe improved flatness for semi-supervised methods along adversarial rays because we do not choose our input or weight perturbations adversarially, but rather they are sampled from a predefined set of transformations.

Additionally, we analyze whether the larger optimization steps for the $\Pi$ and MT models translate into higher *diversity* in predictions. We define diversity of a pair of models $w_1, w_2$ as $\text{Diversity}(w_1, w_2) = \frac{1}{N} \sum_{i=1}^{N} \mathbb{1}[y_i^{(w_1)} \neq y_i^{(w_2)}]$, the fraction of test samples where the predicted labels between the two models differ. We found that for the $\Pi$ and MT models, the $\text{Diversity}(w_{170}, w_{180})$ is 7.1% and 6.1% of the test data points respectively, which is much higher than 3.9% in supervised learning. The increased diversity in the predictions of the networks traversed by SGD supports our conjecture that for the $\Pi$ and MT models SGD struggles to converge to a single solution and continues to actively explore the set of plausible solutions until the end of training.

## 3.3 ENSEMBLING AND WEIGHT AVERAGING

In Section 3.2, we observed that the $\Pi$ and MT models continue taking large steps in the weight space at the end of training. Not only are the distances between weights larger, we observe these models

to have higher diversity. In this setting, using the last SGD iterate to perform prediction is not ideal since many solutions explored by SGD are equally accurate but produce different predictions.

**Ensembling.** In Section 3.2 we showed that the diversity in predictions is significantly larger for the $\Pi$ and Mean Teacher models compared to purely supervised learning. The diversity of these iterates suggests that we can achieve greater benefits from ensembling. We use the same CNN architecture and hyper-parameters as in Section 3.2 but extend the training time by doing 5 learning rate cycles of 30 epochs after the normal training ends at epoch 180 (see A.8 and A.9 for details). We sample random pairs of weights $w_1$, $w_2$ from epochs $180, 183, \ldots, 330$ and measure the error reduction from ensembling these pairs of models, $C_{ens} \equiv \frac{1}{2}\text{Err}(w_1) + \frac{1}{2}\text{Err}(w_2) - \text{Err}\left(\text{Ensemble}(w_1, w_2)\right)$. In Figure 2c we visualize $C_{ens}$, against the diversity of the corresponding pair of models. We observe a strong correlation between the diversity in predictions of the constituent models and ensemble performance, and therefore $C_{ens}$ is substantially larger for $\Pi$ and Mean Teacher models. As shown in Izmailov et al. (2018), ensembling can be well approximated by weight averaging if the weights are close by.

**Weight Averaging.** First, we experiment on averaging random pairs of weights at the end of training and analyze the performance with respect to the weight distances. Using the the same pairs from above, we evaluate the performance of the model formed by averaging the pairs of weights, $C_{avg}(w_1, w_2) \equiv \frac{1}{2}\text{Err}(w_1) + \frac{1}{2}\text{Err}(w_2) - \text{Err}\left(\frac{1}{2}w_1 + \frac{1}{2}w_2\right)$. Note that $C_{avg}$ is a proxy for convexity: if $C_{avg}(w_1, w_2) \geq 0$ for any pair of points $w_1$, $w_2$, then by Jensen's inequality the error function is convex (see the left panel of Figure 2). While the error surfaces for neural networks are known to be highly non-convex, they may be approximately convex in the region traversed by SGD late into training (Goodfellow et al., 2015). In fact, in Figure 2b, we find that the error surface of the SGD trajectory is approximately convex due to $C_{avg}(w_1, w_2)$ being mostly positive. Here we also observe that the distances between pairs of weights are much larger for the $\Pi$ and MT models than for the supervised training; and as a result, weight averaging achieves a larger gain for these models.

In Section 3.2 we observed that for the $\Pi$ and Mean Teacher models SGD traverses a large flat region of the weight space late in training. Being very high-dimensional, this set has most of its volume concentrated near its boundary. Thus, we find SGD iterates at the periphery of this flat region (see Figure 2d). We can also explain this behavior via the argument of (Mandt et al., 2017). Under certain assumptions SGD iterates can be thought of as samples from a Gaussian distribution centered at the minimum of the loss, and samples from high-dimensional Gaussians are known to be concentrated on the surface of an ellipse and never be close to the mean. Averaging the SGD iterates (shown in red in Figure 2d) we can move towards the center (shown in blue) of the flat region, stabilizing the SGD trajectory and improving the width of the resulting solution, and consequently improving generalization.

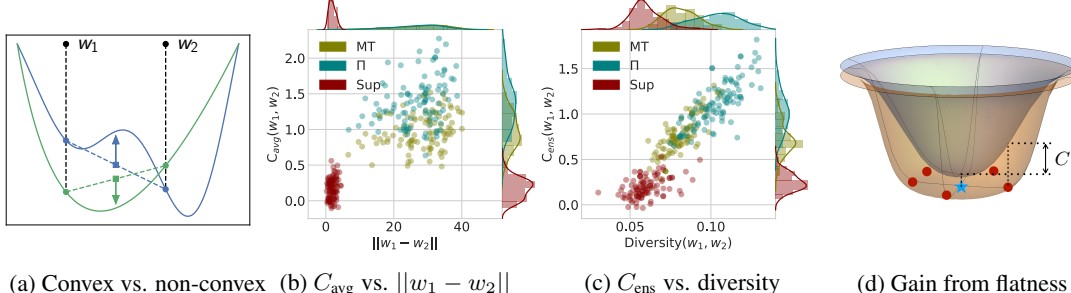

(a) Convex vs. non-convex    (b) $C_{avg}$ vs. $||w_1 - w_2||$      (c) $C_{ens}$ vs. diversity      (d) Gain from flatness

Figure 2: **(a)**: Illustration of a convex and non-convex function and Jensen's inequality. **(b)**: Scatter plot of the decrease in error $C_{avg}$ for weight averaging versus distance. **(c)**: Scatter plot of the decrease in error $C_{ens}$ for prediction ensembling versus diversity. **(d)**: Train error surface (orange) and Test error surface (blue). The SGD solutions (red dots) around a locally flat minimum are far apart due to the flatness of the train surface (see Figure 1b) which leads to large error reduction of the SWA solution (blue dot).

We observe that the improvement $C_{avg}$ from weight averaging ($1.2 \pm 0.2\%$ over MT and $\Pi$ pairs) is on par or larger than the benefit $C_{ens}$ of prediction ensembling ($0.9 \pm 0.2\%$) The smaller gain from

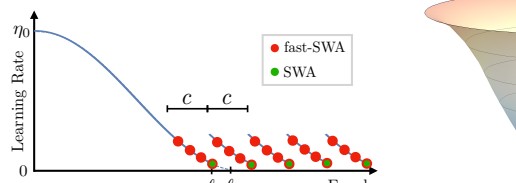 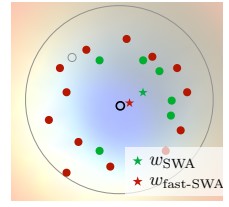

Figure 3: **Left**: Cyclical cosine learning rate schedule and SWA and fast-SWA averaging strategies. **Middle**: Illustration of the solutions explored by the cyclical cosine annealing schedule on an error surface. **Right**: Illustration of SWA and fast-SWA averaging strategies. fast-SWA averages more points but the errors of the averaged points, as indicated by the heat color, are higher.

ensembling might be due to the dependency of the ensembled solutions, since they are from the same SGD run as opposed to independent restarts as in typical ensembling settings. For the rest of the paper, we focus attention on weight averaging because of its lower costs at test time and slightly higher performance compared to ensembling.

## 4 SWA AND FAST-SWA

In Section 3 we analyzed the training trajectories of the $\Pi$, MT, and supervised models. We observed that the $\Pi$ and MT models continue to actively explore the set of plausible solutions, producing diverse predictions on the test set even in the late stages of training. Further, in section 3.3 we have seen that averaging weights leads to significant gains in performance for the $\Pi$ and MT models. In particular these gains are much larger than in supervised setting.

Stochastic Weight Averaging (SWA) (Izmailov et al., 2018) is a recent approach that is based on averaging weights traversed by SGD with a modified learning rate schedule. In Section 3 we analyzed averaging pairs of weights corresponding to different epochs of training and showed that it improves the test accuracy. Averaging multiple weights reinforces this effect, and SWA was shown to significantly improve generalization performance in supervised learning. Based on our results in section 3.3, we can expect even larger improvements in generalization when applying SWA to the $\Pi$ and MT models.

SWA typically starts from a pre-trained model, and then averages points in weight space traversed by SGD with a constant or cyclical learning rate. We illustrate the cyclical cosine learning rate schedule in Figure 3 (left) and the SGD solutions explored in Figure 3 (middle). For the first $\ell \leq \ell_0$ epochs the network is pre-trained using the cosine annealing schedule where the learning rate at epoch $i$ is set equal to $\eta(i) = 0.5 \cdot \eta_0(1 + \cos(\pi \cdot i/\ell_0))$. After $\ell$ epochs, we use a cyclical schedule, repeating the learning rates from epochs $[\ell - c, \ell]$, where $c$ is the cycle length. SWA collects the networks corresponding to the minimum values of the learning rate (shown in green in Figure 3, left) and averages their weights. The model with the averaged weights $w_{\text{SWA}}$ is then used to make predictions. We propose to apply SWA to the student network both for the $\Pi$ and Mean Teacher models. Note that the SWA weights do not interfere with training.

Originally, Izmailov et al. (2018) proposed using cyclical learning rates with small cycle length for SWA. However, as we have seen in Section 3.3 (Figure 2, left) the benefits of averaging are the most prominent when the distance between the averaged points is large. Motivated by this observation, we instead use longer learning rate cycles $c$. Moreover, SWA updates the average weights only once per cycle, which means that many additional training epochs are needed in order to collect enough weights for averaging. To overcome this limitation, we propose **fast-SWA**, a modification of SWA that averages networks corresponding to every $k < c$ epochs starting from epoch $\ell - c$. We can also average multiple weights within a single epoch setting $k < 1$.

Notice that most of the models included in the fast-SWA average (shown in red in Figure 3, left) have higher errors than those included in the SWA average (shown in green in Figure 3, right) since they are obtained when the learning rate is high. It is our contention that including more models in the fast-SWA weight average can more than compensate for the larger errors of the individual models.

Indeed, our experiments in Section 5 show that fast-SWA converges substantially faster than SWA and has lower performance variance. We analyze this result theoretically in Section A.7).

## 5 EXPERIMENTS

We evaluate the $\Pi$ and MT models (Section 4) on CIFAR-10 and CIFAR-100 with varying numbers of labeled examples. We show that fast-SWA and SWA improve the performance of the $\Pi$ and MT models, as we expect from our observations in Section 3. In fact, in many cases fast-SWA improves on the best results reported in the semi-supervised literature. We also demonstrate that the preposed fast-SWA obtains high performance much faster than SWA. We also evaluate SWA applied to a consistency-based domain adaptation model (French et al., 2018), closely related to the MT model, for adapting CIFAR-10 to STL. We improve the best reported test error rate for this task from 19.9% to 16.8%.

We discuss the experimental setup in Section 5.1. We provide the results for CIFAR-10 and CIFAR-100 datasets in Section 5.2 and 5.3. We summarize our results in comparison to the best previous results in Section 5.4. We show several additional results and detailed comparisons in Appendix A.2. We provide analysis of train and test error surfaces of fast-SWA solutions along the directions connecting fast-SWA and SGD in Section A.1.

### 5.1 SETUP

We evaluate the weight averaging methods SWA and fast-SWA on different network architectures and learning rate schedules. We are able to improve on the base models in all settings. In particular, we consider a 13-layer CNN and a 12-block (26-layer) Residual Network (He et al., 2015) with Shake-Shake regularization (Gastaldi, 2017), which we refer to simply as *CNN* and *Shake-Shake* respectively (see Section A.8 for details on the architectures). For training all methods we use the stochastic gradient descent (SGD) optimizer with the cosine annealing learning rate described in Section 4. We use two learning rate schedules, the *short schedule* with $\ell = 180, \ell_0 = 210, c = 30$, similar to the experiments in Tarvainen and Valpola (2017), and the *long schedule* with $\ell = 1500, \ell_0 = 1800, c = 200$, similar to the experiments in Gastaldi (2017). We note that the long schedule improves the performance of the base models compared to the short schedule; however, SWA can still further improve the results. See Section A.9 of the Appendix for more details on other hyperparameters. We repeat each CNN experiment 3 times with different random seeds to estimate the standard deviations for the results in the Appendix.

### 5.2 CIFAR-10

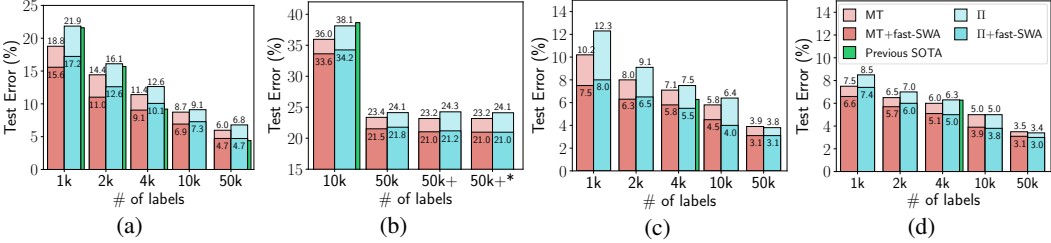

Figure 4: Prediction errors of $\Pi$ and MT models with and without fast-SWA. **(a)** CIFAR-10 with CNN **(b)** CIFAR-100 with CNN. $50k+$ and $50k+^*$ correspond to $50k+500k$ and $50k+237k^*$ settings **(c)** CIFAR-10 with ResNet + Shake-Shake using the short schedule **(d)** CIFAR-10 with ResNet + Shake-Shake using the long schedule.

We evaluate the proposed fast-SWA method using the $\Pi$ and MT models on the CIFAR-10 dataset (Krizhevsky). We use $50k$ images for training with $1k, 2k, 4k, 10k$ and $50k$ labels and report the top-1 errors on the test set ($10k$ images). We visualize the results for the CNN and Shake-Shake architectures in Figures 4a, 4c, and 4d. For all quantities of labeled data, fast-SWA substantially improves test accuracy in both architectures. Additionally, in Tables 2, 4 of the Appendix we provide

a thorough comparison of different averaging strategies as well as results for VAT (Miyato et al., 2017), TE (Laine and Aila, 2016), and other baselines.

Note that we applied fast-SWA for VAT as well which is another popular approach for semi-supervised learning. We found that the improvement on VAT is not drastic – our base implementation obtains $11.26\%$ error where fast-SWA reduces it to $10.97\%$ (see Table 2 in Section A.2). It is possible that the solutions explored by VAT are not as diverse as in $\Pi$ and MT models due to VAT loss function. Throughout the experiments, we focus on the $\Pi$ and MT models as they have been shown to scale to powerful networks such as Shake-Shake and obtained previous state-of-the-art performance.

In Figure 5 (left), we visualize the test error as a function of iteration using the CNN. We observe that when the cyclical learning rate starts after epoch $\ell = 180$, the base models drop in performance due to the sudden increase in learning rate (see Figure 3 left). However, fast-SWA continues to improve while collecting the weights corresponding to high learning rates for averaging. In general, we also find that the cyclical learning rate improves the base models beyond the usual cosine annealing schedule and increases the performance of fast-SWA as training progresses. Compared to SWA, we also observe that fast-SWA converges substantially faster, for instance, reducing the error to $10.5\%$ at epoch 200 while SWA attains similar error at epoch 350 for CIFAR-10 $4k$ labels (Figure 5 left). We provide additional plots in Section A.2 showing the convergence of the $\Pi$ and MT models in all label settings, where we observe similar trends that fast-SWA results in faster error reduction.

We also find that the performance gains of fast-SWA over base models are higher for the $\Pi$ model compared to the MT model, which is consistent with the convexity observation in Section 3.3 and Figure 2. In the previous evaluations (see e.g. Oliver et al., 2018; Tarvainen and Valpola, 2017), the $\Pi$ model was shown to be inferior to the MT model. However, with weight averaging, fast-SWA reduces the gap between $\Pi$ and MT performance. Surprisingly, we find that the $\Pi$ model can outperform MT after applying fast-SWA with moderate to large numbers of labeled points. In particular, the $\Pi$+fast-SWA model outperforms MT+fast-SWA on CIFAR-10 with $4k$, $10k$, and $50k$ labeled data points for the Shake-Shake architecture.

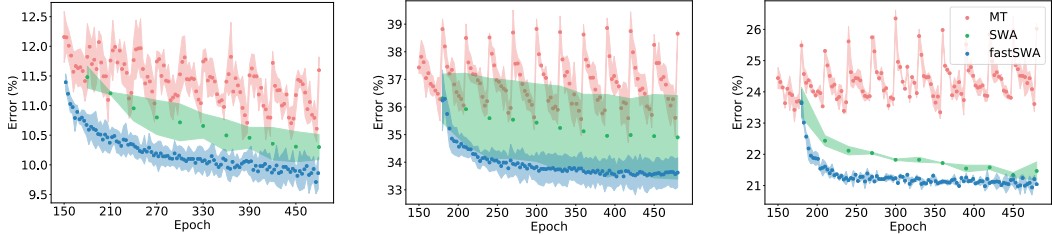

Figure 5: Prediction errors of base models and their weight averages (fast-SWA and SWA) for CNN on **(left)** CIFAR-10 with $4k$ labels, **(middle)** CIFAR-100 with $10k$ labels, and **(right)** CIFAR-100 $50k$ labels and extra $500k$ unlabeled data from Tiny Images (Torralba et al., 2008).

## 5.3 CIFAR-100 AND EXTRA UNLABELED DATA

We evaluate the $\Pi$ and MT models with fast-SWA on CIFAR-100. We train our models using $50000$ images with $10k$ and $50k$ labels using the 13-layer CNN. We also analyze the effect of using the Tiny Images dataset (Torralba et al., 2008) as an additional source of unlabeled data.

The Tiny Images dataset consists of 80 million images, mostly unlabeled, and contains CIFAR-100 as a subset. Following Laine and Aila (2016), we use two settings of unlabeled data, $50k+500k$ and $50k+237k^*$ where the $50k$ images corresponds to CIFAR-100 images from the training set and the $+500k$ or $+237k^*$ images corresponds to additional $500k$ or $237k$ images from the Tiny Images dataset. For the $237k^*$ setting, we select only the images that belong to the classes in CIFAR-100, corresponding to $237203$ images. For the $500k$ setting, we use a random set of $500k$ images whose classes can be different from CIFAR-100. We visualize the results in Figure 4b, where we again observe that fast-SWA substantially improves performance for every configuration of the number of labeled and unlabeled data. In Figure 5 (middle, right) we show the errors of MT, SWA and fast-SWA as a function of iteration on CIFAR-100 for the $10k$ and $50k+500k$ label settings. Similar to the CIFAR-10 experiments, we observe that fast-SWA reduces the errors substantially faster than SWA.

We provide detailed experimental results in Table 3 of the Appendix and include preliminary results using the Shake-Shake architecture in Table 5, Section A.2.

## 5.4 Advancing State-of-the-Art

We have shown that fast-SWA can significantly improve the performance of both the $\Pi$ and MT models. We provide a summary comparing our results with the previous best results in the literature in Table 1, using the 13-layer CNN and the Shake-Shake architecture that had been applied previously. We also provide detailed results the Appendix A.2.

Table 1: Test errors against current state-of-the-art semi-supervised results. The previous best numbers are obtained from (Tarvainen and Valpola, 2017)[1], (Park et al., 2017)[2], (Laine and Aila, 2016)[3] and (Luo et al., 2018)[4]. CNN denotes performance on the benchmark 13-layer CNN (see A.8). Rows marked [†] use the Shake-Shake architecture. The result marked [‡] are from $\Pi$ + fast-SWA, where the rest are based on MT + fast-SWA. The settings $50k+500k$ and $50k+237k^*$ use additional $500k$ and $237k$ unlabeled data from the Tiny Images dataset (Torralba et al., 2008) where $^*$ denotes that we use only the images that correspond to CIFAR-100 classes.

| Dataset | CIFAR-10 | | | CIFAR-100 | | |
|---|---|---|---|---|---|---|
| No. of Images | 50k | 50k | 50k | 50k | 50k+500k | 50k+237k* |
| No. of Labels | 1k | 2k | 4k | 10k | 50k | 50k |
| Previous Best CNN | 18.41[4] | 13.64[4] | 9.22[2] | 38.65[3] | 23.62[3] | 23.79[3] |
| Ours CNN | 15.58 | 11.02 | 9.05 | 33.62 | 21.04 | 20.98 |
| Previous Best[†] | | | 6.28[1] | | | |
| Ours[†] | 6.6 | 5.7 | 5.0[‡] | 28.0 | 19.3 | 17.7 |

## 5.5 Preliminary Results on Domain Adaptation

Domain adaptation problems involve learning using a source domain $X_s$ equipped with labels $Y_s$ and performing classification on the target domain $X_t$ while having no access to the target labels at training time. A recent model by French et al. (2018) applies the consistency enforcing principle for domain adaptation and achieves state-of-the-art results on many datasets. Applying fast-SWA to this model on domain adaptation from CIFAR-10 to STL we were able to improve the best results reported in the literature from $19.9\%$ to $16.8\%$. See Section A.10 for more details on the domain adaptation experiments.

## 6 Discussion

Semi-supervised learning is crucial for reducing the dependency of deep learning on large labeled datasets. Recently, there have been great advances in semi-supervised learning, with consistency regularization models achieving the best known results. By analyzing solutions along the training trajectories for two of the most successful models in this class, the $\Pi$ and Mean Teacher models, we have seen that rather than converging to a single solution SGD continues to explore a diverse set of plausible solutions late into training. As a result, we can expect that averaging predictions or weights will lead to much larger gains in performance than for supervised training. Indeed, applying a variant of the recently proposed stochastic weight averaging (SWA) we advance the best known semi-supervised results on classification benchmarks.

While not the focus of our paper, we have also shown that weight averaging has great promise in domain adaptation (French et al., 2018). We believe that application-specific analysis of the geometric properties of the training objective and optimization trajectories will further improve results over a wide range of application specific areas, including reinforcement learning with sparse rewards, generative adversarial networks (Yazıcı et al., 2018), or semi-supervised natural language processing.

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

# A APPENDIX

## A.1 ADDITIONAL PLOTS

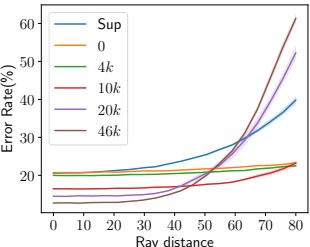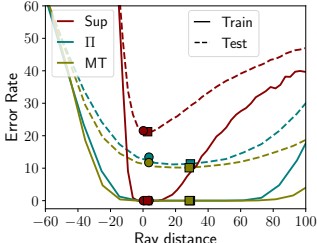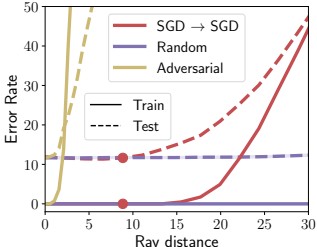

Figure 6: All plots are a obtained using the 13-layer CNN on CIFAR-10 with $4k$ labeled and $46k$ unlabeled data points unless specified otherwise. **Left**: Test error as a function of distance along random rays for the $\Pi$ model with $0$, $4k$, $10k$, $20k$ or $46k$ unlabeled data points, and standard fully supervised training which uses only the cross entropy loss. All methods use $4k$ labeled examples. **Middle**: Train and test errors along rays connecting SGD solutions (showed with circles) to SWA solutions (showed with squares) for each respective model. **Right**: Comparison of train and test errors along rays connecting two SGD solutions, random rays, and adversarial rays for the Mean Teacher model.

In this section we provide several additional plots visualizing the train and test error along different types of rays in the weight space. The left panel of Figure 6 shows how the behavior of test error changes as we add more unlabeled data points for the $\Pi$ model. We observe that the test accuracy improves monotonically, but also the solutions become narrower along random rays.

The middle panel of Figure 6 visualizes the train and test error behavior along the directions connecting the fast-SWA solution (shown with squares) to one of the SGD iterates used to compute the average (shown with circles) for $\Pi$, MT and supervised training. Similarly to Izmailov et al. (2018) we observe that for all three methods fast-SWA finds a centered solution, while the SGD solution lies near the boundary of a wide flat region. Agreeing with our results in section 3.2 we observe that for $\Pi$ and Mean Teacher models the train and test error surfaces are much wider along the directions connecting the fast-SWA and SGD solutions than for supervised training.

In the right panel of Figure 6 we show the behavior of train and test error surfaces along random rays, adversarial rays and directions connecting the SGD solutions from epochs 170 and 180 for the Mean Teacher model (see section 3.2).

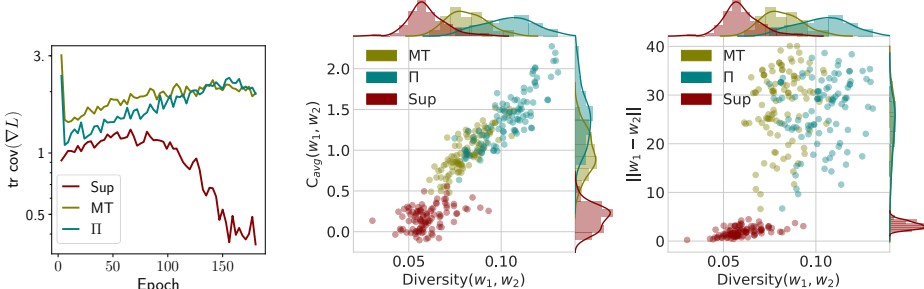

Figure 7: (**Left**): The evolution of the gradient covariance trace in the $\Pi$, MT, and supervised models during training. (**Middle**): Scatter plot of the decrease in error $C_{avg}$ for weight averaging versus diversity. (**Right**): Scatter plot of the distance between pairs of weights versus diversity in their predictions.

In the left panel of Figure 7 we show the evolution of the trace of the gradient of the covariance of the loss

$$\operatorname{tr} \operatorname{cov}(\nabla_w L(w)) = \mathbb{E}\|\nabla_w L(w) - \mathbb{E}\nabla_w L(w)\|^2$$

for the $\Pi$, MT and supevised training. We observe that the variance of the gradient is much larger for the $\Pi$ and Mean Teacher models compared to supervised training.

In the middle and right panels of figure 7 we provide scatter plots of the improvement $C$ obtained from averaging weights against diversity and diversity against distance. We observe that diversity is highly correlated with the improvement $C$ coming from weight averaging. The correlation between distance and diversity is less prominent.

## A.2    DETAILED RESULTS

In this section we report detailed results for the $\Pi$ and Mean Teacher models and various baselines on CIFAR-10 and CIFAR-100 using the 13-layer CNN and Shake-Shake.

The results using the 13-layer CNN are summarized in Tables 2 and 3 for CIFAR-10 and CIFAR-100 respectively. Tables 4 and 5 summarize the results using Shake-Shake on CIFAR-10 and CIFAR-100. In the tables $\Pi$ EMA is the same method as $\Pi$, where instead of SWA we apply Exponential Moving Averaging (EMA) for the student weights. We show that simply performing EMA for the student network in the $\Pi$ model without using it as a teacher (as in MT) typically results in a small improvement in the test error.

Figures 8 and 9 show the performance of the $\Pi$ and Mean Teacher models as a function of the training epoch for CIFAR-10 and CIFAR-100 respectively for SWA and fast-SWA.

Table 2: CIFAR-10 semi-supervised errors on test set with a 13-layer CNN. The epoch numbers are reported in parenthesis. The previous results shown in the first section of the table are obtained from Tarvainen and Valpola (2017)[1], Park et al. (2017)[2], Laine and Aila (2016)[3], Miyato et al. (2017)[4].

| Number of labels | 1000 | 2000 | 4000 | 10000 | 50000 |
|---|---|---|---|---|---|
| TE[3] | - | - | $12.16 \pm 0.31$ | | $5.60 \pm 0.15$ |
| Supervised-only[1] | $46.43 \pm 1.21$ | $33.94 \pm 0.73$ | $20.66 \pm 0.57$ | | $5.82 \pm 0.15$ |
| $\Pi$ [1] | $27.36 \pm 1.20$ | $18.02 \pm 0.60$ | $13.20 \pm 0.27$ | | $6.06 \pm 0.15$ |
| MT[1] | $21.55 \pm 1.48$ | $15.73 \pm 0.31$ | $12.31 \pm 0.28$ | | $5.94 \pm 0.15$ |
| VAdD[3] | | | $9.22 \pm 0.10$ | | $\mathbf{4.40 \pm 0.12}$ |
| VAT + EntMin[4] | | | $10.55$ | | |
| MT | $18.78 \pm 0.31$ | $14.43 \pm 0.20$ | $11.41 \pm 0.27$ | $8.74 \pm 0.30$ | $5.98 \pm 0.21$ |
| MT + fast-SWA (180) | $18.19 \pm 0.38$ | $13.46 \pm 0.30$ | $10.67 \pm 0.18$ | $8.06 \pm 0.12$ | $5.90 \pm 0.03$ |
| MT + fast-SWA (240) | $17.81 \pm 0.37$ | $13.00 \pm 0.31$ | $10.34 \pm 0.14$ | $7.73 \pm 0.10$ | $5.55 \pm 0.03$ |
| MT + SWA (240) | $18.38 \pm 0.29$ | $13.86 \pm 0.64$ | $10.95 \pm 0.21$ | $8.36 \pm 0.50$ | $5.75 \pm 0.29$ |
| MT + fast-SWA (480) | $16.84 \pm 0.62$ | $12.24 \pm 0.31$ | $9.86 \pm 0.27$ | $7.39 \pm 0.14$ | $5.14 \pm 0.07$ |
| MT + SWA (480) | $17.48 \pm 0.13$ | $13.09 \pm 0.80$ | $10.30 \pm 0.21$ | $7.78 \pm 0.49$ | $5.31 \pm 0.43$ |
| MT + fast-SWA (1200) | $\mathbf{15.58 \pm 0.12}$ | $\mathbf{11.02 \pm 0.23}$ | $\mathbf{9.05 \pm 0.21}$ | $\mathbf{6.92 \pm 0.07}$ | $\mathbf{4.73 \pm 0.18}$ |
| MT + SWA (1200) | $15.59 \pm 0.77$ | $11.42 \pm 0.33$ | $9.38 \pm 0.28$ | $7.04 \pm 0.11$ | $5.11 \pm 0.35$ |
| $\Pi$ | $21.85 \pm 0.69$ | $16.10 \pm 0.51$ | $12.64 \pm 0.11$ | $9.11 \pm 0.21$ | $6.79 \pm 0.22$ |
| $\Pi$ EMA | $21.70 \pm 0.57$ | $15.83 \pm 0.55$ | $12.52 \pm 0.16$ | $9.06 \pm 0.15$ | $6.66 \pm 0.20$ |
| $\Pi$ + fast-SWA (180) | $20.79 \pm 0.38$ | $15.12 \pm 0.44$ | $11.91 \pm 0.06$ | $8.83 \pm 0.32$ | $6.42 \pm 0.09$ |
| $\Pi$ + fast-SWA (240) | $20.04 \pm 0.41$ | $14.77 \pm 0.15$ | $11.61 \pm 0.06$ | $8.45 \pm 0.28$ | $6.14 \pm 0.11$ |
| $\Pi$ + SWA (240) | $21.37 \pm 0.64$ | $15.38 \pm 0.85$ | $12.05 \pm 0.40$ | $8.58 \pm 0.41$ | $6.36 \pm 0.55$ |
| $\Pi$ + fast-SWA (480) | $19.11 \pm 0.29$ | $13.88 \pm 0.30$ | $10.91 \pm 0.15$ | $7.91 \pm 0.21$ | $5.53 \pm 0.07$ |
| $\Pi$ + SWA (480) | $20.06 \pm 0.64$ | $14.53 \pm 0.81$ | $11.35 \pm 0.42$ | $8.04 \pm 0.37$ | $5.77 \pm 0.51$ |
| $\Pi$ + fast-SWA (1200) | $\mathbf{17.23 \pm 0.34}$ | $\mathbf{12.61 \pm 0.18}$ | $\mathbf{10.07 \pm 0.27}$ | $\mathbf{7.28 \pm 0.23}$ | $\mathbf{4.72 \pm 0.04}$ |
| $\Pi$ + SWA (1200) | $17.70 \pm 0.25$ | $12.59 \pm 0.29$ | $10.73 \pm 0.39$ | $7.13 \pm 0.23$ | $4.99 \pm 0.41$ |
| VAT | | | $11.99$ | | |
| VAT + SWA | | | $11.16$ | | |
| VAT+ EntMin | | | $11.26$ | | |
| VAT + EntMin + SWA | | | $\mathbf{10.97}$ | | |

Table 3: CIFAR-100 semi-supervised errors on test set. All models are trained on a 13-layer CNN. The epoch numbers are reported in parenthesis. The previous results shown in the first section of the table are obtained from (Laine and Aila, 2016)[3].

| Number of labels | 10k | 50k | 50k + 500k | 50k + 237k* |
|---|---|---|---|---|
| Supervised-only[3] | $44.56 \pm 0.30$ | $26.42 \pm 0.17$ | | |
| Π model[3] | $39.19 \pm 0.54$ | $26.32 \pm 0.04$ | $25.79 \pm 0.17$ | $25.43 \pm 0.17$ |
| Temporal Ensembling[3] | $38.65 \pm 0.51$ | $26.30 \pm 0.15$ | $23.62 \pm 0.17$ | $23.79 \pm 0.17$ |
| MT (180) | $35.96 \pm 0.77$ | $23.37 \pm 0.16$ | $23.18 \pm 0.06$ | $23.18 \pm 0.24$ |
| MT + fast-SWA (180) | $34.54 \pm 0.48$ | $21.93 \pm 0.16$ | $21.04 \pm 0.16$ | $21.09 \pm 0.12$ |
| MT + SWA (240) | $35.59 \pm 1.45$ | $23.17 \pm 0.86$ | $22.00 \pm 0.23$ | $21.59 \pm 0.22$ |
| MT + fast-SWA (240) | $34.10 \pm 0.31$ | $21.84 \pm 0.12$ | $21.16 \pm 0.21$ | $21.07 \pm 0.21$ |
| MT + SWA (1200) | $34.90 \pm 1.51$ | $22.58 \pm 0.79$ | $21.47 \pm 0.29$ | $21.27 \pm 0.09$ |
| MT + fast-SWA (1200) | $\mathbf{33.62 \pm 0.54}$ | $\mathbf{21.52 \pm 0.12}$ | $\mathbf{21.04 \pm 0.04}$ | $\mathbf{20.98 \pm 0.36}$ |
| Π (180) | $38.13 \pm 0.52$ | $24.13 \pm 0.20$ | $24.26 \pm 0.15$ | $24.10 \pm 0.07$ |
| Π + fast-SWA (180) | $35.59 \pm 0.62$ | $22.08 \pm 0.21$ | $21.40 \pm 0.19$ | $21.28 \pm 0.20$ |
| Π + SWA (240) | $36.89 \pm 1.51$ | $23.23 \pm 0.70$ | $22.17 \pm 0.19$ | $21.65 \pm 0.13$ |
| Π + fast-SWA (240) | $35.14 \pm 0.71$ | $22.00 \pm 0.21$ | $21.29 \pm 0.27$ | $21.22 \pm 0.04$ |
| Π + SWA (1200) | $35.35 \pm 1.15$ | $22.53 \pm 0.64$ | $21.53 \pm 0.13$ | $21.26 \pm 0.34$ |
| Π + fast-SWA (1200) | $\mathbf{34.25 \pm 0.16}$ | $\mathbf{21.78 \pm 0.05}$ | $\mathbf{21.19 \pm 0.05}$ | $\mathbf{20.97 \pm 0.08}$ |

Table 4: CIFAR-10 semi-supervised errors on test set. All models use Shake-Shake Regularization (Gastaldi, 2017) + ResNet-26 (He et al., 2015).

| Number of labels | 1000 | 2000 | 4000 | 10000 | 50000 |
|---|---|---|---|---|---|
| Short Schedule ($\ell = 180$) | | | | | |
| MT[†] (Tarvainen and Valpola, 2017) | | | 6.28 | | |
| MT (180) | 10.2 | 8.0 | 7.1 | 5.8 | 3.9 |
| MT + SWA (240) | 9.7 | 7.7 | 6.2 | 4.9 | 3.4 |
| MT + fast-SWA (240) | 9.6 | 7.4 | 6.2 | 4.9 | 3.2 |
| MT + SWA (1200) | 7.6 | 6.4 | 5.8 | 4.6 | **3.1** |
| MT + fast-SWA (1200) | **7.5** | **6.3** | 5.8 | 4.5 | **3.1** |
| Π (180) | 12.3 | 9.1 | 7.5 | 6.4 | 3.8 |
| Π + SWA (240) | 11.0 | 8.3 | 6.7 | 5.5 | 3.3 |
| Π + fast-SWA (240) | 11.2 | 8.2 | 6.7 | 5.5 | 3.3 |
| Π + SWA (1200) | 8.2 | 6.7 | 5.7 | 4.2 | **3.1** |
| Π + fast-SWA (1200) | 8.0 | 6.5 | **5.5** | **4.0** | **3.1** |
| Long Schedule ($\ell = 1500$) | | | | | |
| Supervised-only (Gastaldi, 2017) | | | | | **2.86** |
| MT (1500) | 7.5 | 6.5 | 6.0 | 5.0 | 3.5 |
| MT + fast-SWA (1700) | 6.4 | 5.8 | 5.2 | 3.8 | 3.4 |
| MT + SWA (1700) | 6.9 | 5.9 | 5.5 | 4.2 | 3.2 |
| MT + fast-SWA (3500) | **6.6** | **5.7** | 5.1 | 3.9 | 3.1 |
| MT + SWA (3500) | 6.7 | 5.8 | 5.2 | 3.9 | 3.1 |
| Π (1500) | 8.5 | 7.0 | 6.3 | 5.0 | 3.4 |
| Π + fast-SWA (1700) | 7.5 | 6.2 | 5.2 | 4.0 | 3.1 |
| Π + SWA (1700) | 7.8 | 6.4 | 5.6 | 4.4 | 3.2 |
| Π + fast-SWA (3500) | 7.4 | 6.0 | **5.0** | **3.8** | 3.0 |
| Π + SWA (3500) | 7.9 | 6.2 | 5.1 | 4.0 | 3.0 |

Table 5: CIFAR-100 semi-supervised errors on test set. Our models use Shake-Shake Regularization (Gastaldi, 2017) + ResNet-26 (He et al., 2015).

| Number of labels | 10k | 50k | 50k + 500k | 50k + 237k* |
|---|---|---|---|---|
| TE (CNN) (Laine and Aila, 2016) | $38.65 \pm 0.51$ | $26.30 \pm 0.15$ | $23.62 \pm 0.17$ | $23.79 \pm 0.17$ |
| Short Schedule ($\ell = 180$) | | | | |
| MT (180) | 29.4 | 19.5 | 21.9 | 19.0 |
| MT + fast-SWA (180) | 28.9 | 19.3 | 19.7 | 18.3 |
| MT + SWA (240) | 28.4 | 18.8 | 19.9 | 17.9 |
| MT + fast-SWA (240) | 28.1 | 18.8 | 19.5 | 17.9 |
| MT + SWA (300) | 28.1 | 18.5 | **18.9** | **17.5** |
| MT + fast-SWA (300) | **28.0** | **18.4** | 19.3 | 17.7 |

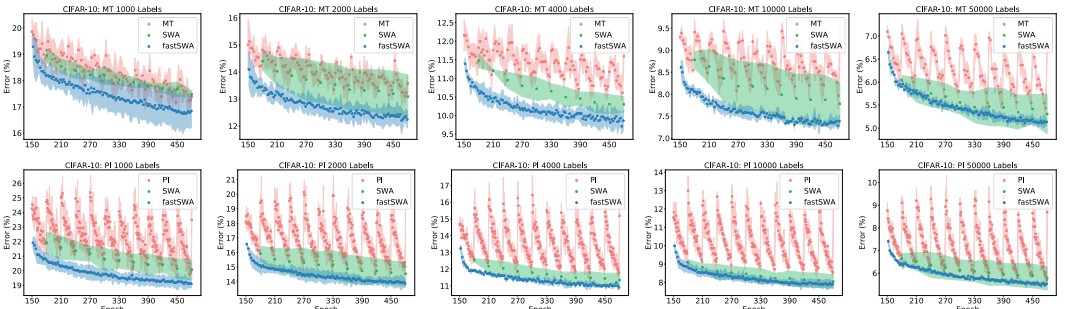

Figure 8: Test errors as a function of training epoch for baseline models, SWA and fast-SWA on CIFAR-10 trained using $1k$, $2k$, $4k$, and $10k$ labels for **(top)** the MT model **(bottom)** the $\Pi$ model. All models are trained using the 13-layer CNN.

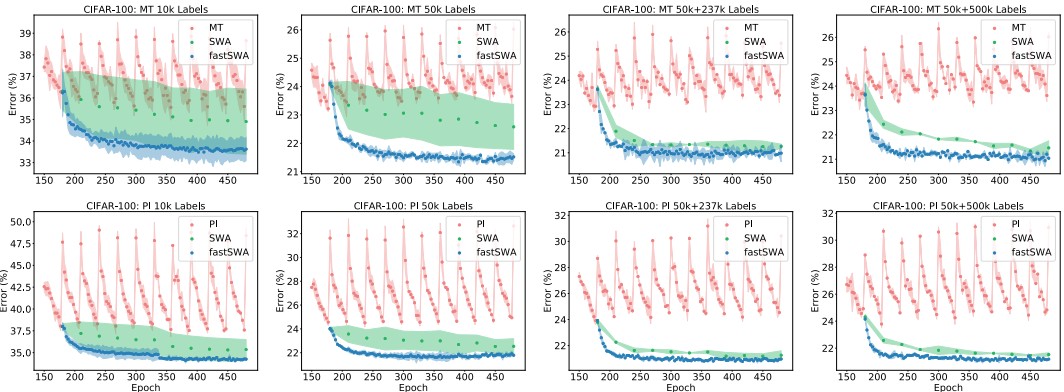

Figure 9: Test errors versus training epoch for baseline models, SWA and fast-SWA on CIFAR-100 trained using $10k$, $50k$, $50k+500k$, and $50k+237k^*$ labels for **(top)** the MT model **(bottom)** the $\Pi$ model. All models are trained using the 13-layer CNN.

## A.3   EFFECT OF LEARNING RATE SCHEDULES

The only hyperparameter for the fast-SWA setting is the cycle length $c$. We demonstrate in Figure 10a that fast-SWA's performance is not sensitive to $c$ over a wide range of $c$ values. We also demonstrate the performance for constant learning schedule. fast-SWA with cyclical learning rates generally converges faster due to higher variety in the collected weights.

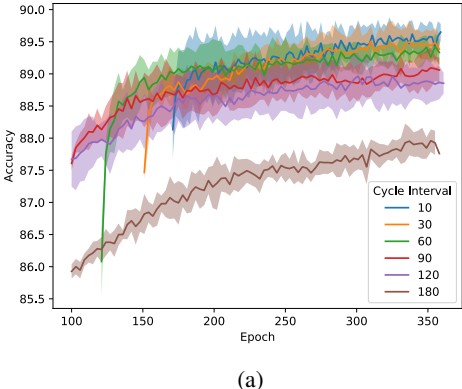 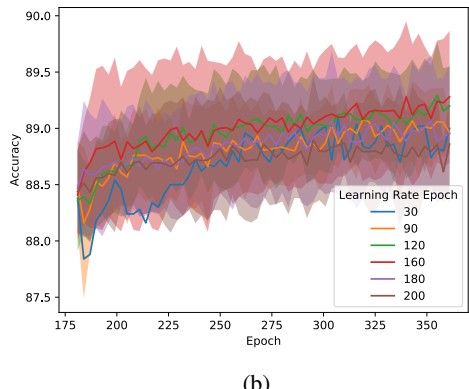

(a)                  (b)

Figure 10: The plots are generated using the MT model with CNN trained on CIFAR-10. We randomly select $5k$ of the $50k$ train images as a validation set. The remaining $45k$ images are splitted into $4k$ labeled and $41k$ unlabeled data points. **(a)** Validation accuracy as a function of training epoch for different cycle lengths $c$ **(b)** fast-SWA with constant learning rate. The "learning rate epoch" corresponds to the epoch in the unmodified cosine annealing schedule (Figure 3, left) at which the learning rate is evaluated. We use this fixed learning rate for all epochs $i \geq \ell$.

## A.4 EMA VERSUS SWA AS A TEACHER

The MT model uses an exponential moving average (EMA) of the student weights as a teacher in the consistency regularization term. We consider two potential effects of using EMA as a teacher: first, averaging weights improves performance of the teacher for the reasons discussed in Sections 3.2, 3.3; second, having a better teacher model leads to better student performance which in turn further improves the teacher. In this section we try to separate these two effects. We apply EMA to the Π model in the same way in which we apply fast-SWA instead of using EMA as a teacher and compare the resulting performance to the Mean Teacher. Figure 11 shows the improvement in error-rate obtained by applying EMA to the Π model in different label settings. As we can see while EMA improves the results over the baseline Π model, the performance of Π-EMA is still inferior to that of the Mean Teacher method, especially when the labeled data is scarce. This observation suggests that the improvement of the Mean Teacher over the Π model can not be simply attributed to EMA improving the student performance and we should take the second effect discussed above into account.

Like SWA, EMA is a way to average weights of the networks, but it puts more emphasis on very recent models compared to SWA. Early in training when the student model changes rapidly EMA significantly improves performance and helps a lot when used as a teacher. However once the student model converges to the vicinity of the optimum, EMA offers little gain. In this regime SWA is a much better way to average weights. We show the performance of SWA applied to Π model in Figure 11 (left).

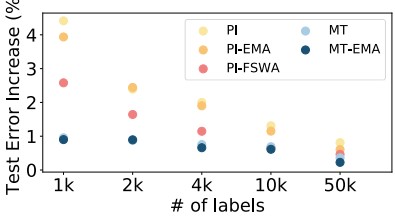 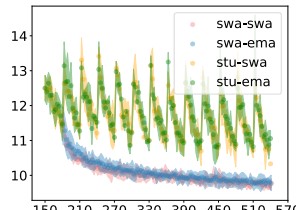

Figure 11: **(left)** Comparison of different averaging methods. The y axis corresponds to the increased error with respect to the MT model with fast-SWA solution (which has $y = 0$). All numbers are taken from epoch 180. **(right)** The effects of using SWA as a teacher. W-T model corresponds to the performance of a model with weight W using a model with a teacher being T.

Since SWA performs better than EMA, we also experiment with using SWA as a teacher instead of EMA. We start with the usual MT model pretrained until epoch 150. Then we switch to using SWA as a teacher at epoch 150. In Figure 11 (right), our results suggest that using SWA as a teacher performs on par with using EMA as a teacher. We conjecture that once we are at a convex region of test error close to the optimum (epoch 150), having a better teacher doesn't lead to substantially improved performance. It is possible to start using SWA as a teacher earlier in training; however, during early epochs where the model undergoes rapid improvement EMA is more sensible than SWA as we discussed above.

### A.5 CONSISTENCY LOSS APPROXIMATES JACOBIAN NORM

**Estimator mean and variance:** In the simplified $\Pi$ model with small additive data perturbations that are normally distributed, $z \sim \mathcal{N}(0, I)$,

$$\hat{Q} = \lim_{\epsilon \to 0} \frac{1}{\epsilon^2} \frac{1}{m} \sum_{i=1}^{m} \ell_{cons}(w, x_i, \epsilon) = \frac{1}{m} \sum_{i=1}^{m} \lim_{\epsilon \to 0} \frac{1}{\epsilon^2} \| f(w, x_i + \epsilon z_i) - f(w, x_i) \|^2$$

Taylor expanding $\ell_{cons}$ in $\epsilon$, we obtain $\ell_{cons}(w, x, \epsilon) = \epsilon^2 z^T J_x^T J_x z + O(\epsilon^4)$, where $J_x$ is the Jacobian of the network outputs $f$ with respect to the input at a particular value of $x$. Therefore,

$$\hat{Q}_i = \lim_{\epsilon \to 0} \frac{1}{\epsilon^2} \ell_{cons}(w, x_i, \epsilon) = z_i^T J(x_i)^T J(x_i) z_i.$$

We can now recognize this term as a one sample stochastic trace estimator for $\text{tr}(J(x_i)^T J(x_i))$ with a Gaussian probe variable $z_i$; see Avron and Toledo (2011) for derivations and guarantees on stochastic trace estimators.

$$\mathbb{E}_z[\hat{Q}_i] = \text{tr}\left(J(x_i)^T J(x_i) \mathbb{E}[z_i z_i^T]\right) = \|J(x_i)\|_F^2.$$

Taking an expectation over the $m$ samples of $x$, we get $\mathbb{E}[\hat{Q}] = \mathbb{E}_x[\|J_x\|_F^2]$.

In general if we have $m$ samples of $x$ and $n$ sampled perturbations for each $x$, then for a symmetric matrix $A$ with $z_{ik} \stackrel{iid}{\sim} N(0, I)$ and independent $x_i \stackrel{iid}{\sim} p(x)$,

the estimator

$$\hat{Q} = \frac{1}{m} \sum_i^m \frac{1}{n} \sum_k^n z_{ik}^T A(x_i) z_{ik} \quad \text{has variance}$$

$$\text{Var}[\hat{Q}] = \frac{1}{m} \left( \text{Var}[\text{tr}(A)] + \frac{2}{n} \mathbb{E}[\text{tr}(A^2)] \right).$$

Proof: Let $q_{ik} \equiv z_{ik}^T A(x_i) z_{ik}$. It is easy to show that for fixed $x$, $\mathbb{E}_z[q_{11}|x_1] = 2\text{tr}(A(x_1)^2) + \text{tr}(A(x_1))^2$, (see e.g. Avron and Toledo, 2011). Note that $\mathbb{E}_{z_1, z_2}[q_{i1} q_{i2}|x_i] = \text{tr}(A(x_i))^2$. Since $\left\{ \frac{1}{n} \sum_k^n q_{ik} \right\}_{i=1}^m$ are i.i.d random variables,

$$\text{Var}\left[\frac{1}{m} \sum_i^m \frac{1}{n} \sum_k^n q_{ik}\right] = \frac{1}{m} \text{Var}\left[\frac{1}{n} \sum_k^n q_{1k}\right],$$

whereas this does not hold for the opposite ordering of the sum.

$$\mathbb{E}\left[\left(\frac{1}{n} \sum_k^n q_{1k}\right)^2\right] = \mathbb{E}_{x_1} \mathbb{E}_{\{z\}} \left[\frac{1}{n^2} \sum_l^n \sum_k^n q_{il} q_{ik} \big| \{x\}\right]$$

$$= \mathbb{E}_{x_1} \mathbb{E}_{\{z\}} \left[\frac{n}{n^2} q_{11}^2 + \frac{n(n-1)}{n^2} q_{11} q_{12} \big| \{x\}\right]$$

$$= \mathbb{E}_x \left[\frac{1}{n}\left(2\text{tr}(A^2) + \text{tr}(A)^2\right) + \left(1 - \frac{1}{n}\right)\text{tr}(A)^2\right] = \left(\mathbb{E}_x[\text{tr}(A)^2] + \frac{2}{n}\mathbb{E}_x[\text{tr}(A^2)]\right)$$

Plugging in $A = J^T J$ and $n = 1$, we get

$$\text{Var}[\hat{Q}] = \frac{1}{m}\left(\text{Var}[\|J_x\|_F^2] + 2\mathbb{E}[\|J_x^T J_x\|_F^2]\right).$$

**Non-isotropic perturbations along data manifold** Consistency regularization with natural pertur-
bations such as image translation can also be understood as penalizing a Jacobian norm as in Section
3.1. For example, consider perturbations sampled from a normal distribution on the tangent space,
$z \sim P(x)\mathcal{N}(0, I)$ where $P(x) = P(x)^2$ is the orthogonal projection matrix that projects down from
$\mathbb{R}^d$ to $T_x(\mathcal{M})$, the tangent space of the image manifold at $x$. Then the consistency regularization
penalizes the Laplacian norm of the network on the manifold (with the inherited metric from $\mathbb{R}^d$).
$\mathbb{E}[z] = 0$ and $\mathbb{E}[zz^T] = PP^T(=)P^2 = P$ which follows if $P$ is an orthogonal projection matrix.
Then,
$$\mathbb{E}[z^T J^T J z] = \text{tr}(J^T J P P^T) = \text{tr}(P^T J^T J P) = \text{tr}(J_{\mathcal{M}}^T J_{\mathcal{M}}) = \|J_{\mathcal{M}}\|_F^2.$$
We view the standard data augmentations such as random translation (that are applied in the $\Pi$
and MT models) as approximating samples of nearby elements of the data manifold and therefore
differences $x' - x$ approximate elements of its tangent space.

## A.6 Relationship Between $\mathbb{E}_x[\|J_w\|_F^2]$ and Random Ray Sharpness

In the following analysis we review an argument for why smaller $\mathbb{E}_x[\|J_w\|_F^2]$, implies broader optima.
To keep things simple, we focus on the MSE loss, but in principle a similar argument should apply
for the Cross Entropy and the Error rate. For a single data point $x$ and one hot vector $y$ with $k$ classes,
the hessian of $\ell_{MSE}(w) = \|f(x, w) - y\|^2$ can be decomposed into two terms, the Gauss-Newton
matrix $G = J_w^T J_w$ and a term which depends on the labels.

$$H(w, x, y) = \nabla^2 \ell_{MSE}(w) = J_w^T J_w + \sum_{i=1}^k (\nabla^2 f_i)(f_i(x) - y_i),$$

$$\text{tr}(H) = \|J_w\|_F^2 + \underbrace{\sum_{i=1}^k \text{tr}(\nabla^2 f_i)(f_i(x) - y_i)}_{\alpha(x,y)}$$

Thus $\text{tr}(H)$ is also the sum of two terms, $\|J_w\|_F^2$ and $\alpha$. As the solution improves, the relative
size of $\alpha$ goes down. In terms of random ray sharpness, consider the expected MSE loss, or risk,
$R_{\text{MSE}}(w) = \mathbb{E}_{(x,y)}\|f(x, w) - y\|^2$ along random rays. Let $d$ be a random vector sampled from the
unit sphere and $s$ is the distance along the random ray. Evaluating the risk on a random ray, and
Taylor expanding in $s$ we have

$$R_{\text{MSE}}(w + sd) = R_{\text{MSE}}(w) + sd^T \mathbb{E}_{(x,y)}[J_w^T(f - y)] + (1/2)s^2 d^T \mathbb{E}_{(x,y)}[H]d + O(s^3)$$

Since $d$ is from the unit sphere, $\mathbb{E}[d] = 0$ and $\mathbb{E}[dd^T] = I/p$ where $p$ is the dimension. Averaging
over the rays, $d \sim \text{Unif}(S^{p-1})$, we have

$$\mathbb{E}_d[R_{\text{MSE}}(w+sd)] - R_{\text{MSE}}(w) = \frac{s^2}{2p}\mathbb{E}_x[\text{tr}(H)] + O(s^4) = \frac{s^2}{2p}\mathbb{E}_x[\|J_w\|_F^2] + \frac{s^2}{2p}\mathbb{E}_{(x,y)}[\alpha(x,y)] + O(s^4)$$

All of the odd terms vanish because of the reflection symmetry of the unit sphere. This means that
locally, the sharpness of the optima (as measured by random rays) can be lowered by decreasing
$\mathbb{E}_x[\|J_w\|_F^2]$.

## A.7 Including High Learning Rate Iterates Into SWA

As discussed in Mandt et al. (2017), under certain assumptions SGD samples from a Gaussian
distribution centered at the optimum of the loss $w_0$ with covariance proportional to the learning
rate. Suppose then that we have $n$ weights sampled at learning rate $\eta_1$, $w_i^{(1)} \overset{iid}{\sim} \mathcal{N}(w_0, \eta_1\Sigma)$
and $m$ weights sampled with the higher learning rate $\eta_2$, $w_j^{(2)} \overset{iid}{\sim} \mathcal{N}(w_0, \eta_2\Sigma)$. For the SWA
estimator $\hat{w}_{\text{SWA}} = \frac{1}{n}\sum_i w_i^{(1)}$, $\mathbb{E}[\|\hat{w}_{\text{SWA}} - w_0\|^2] = \text{tr}(\text{Cov}(\hat{w}_{\text{SWA}})) = \frac{\eta_1}{n}\text{tr}(\Sigma)$. But if we include
the high variance points in the average, as in fast-SWA, $\hat{w}_{\text{fSWA}} = \frac{1}{n+m}\left(\sum_i w_i^{(1)} + \sum_j w_j^{(2)}\right)$, then
$\mathbb{E}[\|\hat{w}_{\text{fSWA}} - w_0\|^2] = \frac{n\eta_1+m\eta_2}{(n+m)^2}\text{tr}(\Sigma)$. If $\frac{n\eta_1+m\eta_2}{(n+m)^2} < \frac{\eta_1}{n}$ then including the high learning rate points
decreases the MSE of the estimator for $m > n\left(\frac{\eta_2}{\eta_1} - 2\right)$. If we include enough points, we will still
improve the estimate.

## A.8 NETWORK ARCHITECTURES

In the experiments we use two DNN architectures – 13 layer CNN and Shake-Shake. The architecture of 13-layer CNN is described in Table 6. It closely follows the architecture used in (Laine and Aila, 2017; Miyato et al., 2017; Tarvainen and Valpola, 2017). We re-implement it in PyTorch and removed the Gaussian input noise, since we found having no such noise improves generalization. For Shake-Shake we use 26-2x96d Shake-Shake regularized architecture of Gastaldi (2017) with 12 residual blocks.

Table 6: A 13-layer convolutional neural networks for the CNN experiments (CIFAR-10 and CIFAR-100) in Section 5.2 and 5.3. Note that the difference from the architecture used in Tarvainen and Valpola (2017) is that we removed a Gaussian noise layer after the horizontal flip.

| Layer | Hyperparameters |
| --- | --- |
| Input | $32 \times 32$ RGB image |
| Translation | Randomly $\{\Delta x, \Delta y\} \sim [-4, 4]$ |
| Horizontal flip | Randomly $p = 0.5$ |
| Convolutional | 128 filters, $3 \times 3$, *same* padding |
| Convolutional | 128 filters, $3 \times 3$, *same* padding |
| Convolutional | 128 filters, $3 \times 3$, *same* padding |
| Pooling | Maxpool $2 \times 2$ |
| Dropout | $p = 0.5$ |
| Convolutional | 256 filters, $3 \times 3$, *same* padding |
| Convolutional | 256 filters, $3 \times 3$, *same* padding |
| Convolutional | 256 filters, $3 \times 3$, *same* padding |
| Pooling | Maxpool $2 \times 2$ |
| Dropout | $p = 0.5$ |
| Convolutional | 512 filters, $3 \times 3$, *valid* padding |
| Convolutional | 256 filters, $1 \times 1$, *same* padding |
| Convolutional | 128 filters, $1 \times 1$, *same* padding |
| Pooling | Average pool ($6 \times 6 \rightarrow 1 \times 1$ pixels) |
| Softmax | Fully connected $128 \rightarrow 10$ |

## A.9 HYPERPARAMETERS

We consider two different schedules. In the *short schedule* we set the cosine half-period $\ell_0 = 210$ and training length $\ell = 180$, following the schedule used in Tarvainen and Valpola (2017) in Shake-Shake experiments. For our Shake-Shake experiments we also report results with *long schedule* where we set $\ell = 1800, \ell_0 = 1500$ following Gastaldi (2017). To determine the initial learning rate $\eta_0$ and the cycle length $c$ we used a separate validation set of size 5000 taken from the unlabeled data. After determining these values, we added the validation set to the unlabeled data and trained again. We reuse the same values of $\eta_0$ and $c$ for all experiments with different numbers of labeled data for both $\Pi$ model and Mean Teacher for a fixed architecture (13-layer CNN or Shake-Shake). For the short schedule we use cycle length $c = 30$ and average models once every $k = 3$ epochs. For long schedule we use $c = 200, k = 20$.

In all experiments we use stochastic gradient descent optimizer with Nesterov momentum (Loshchilov and Hutter, 2016). In fast-SWA we average every the weights of the models corresponding to every third epoch. In the $\Pi$ model, we back-propagate the gradients through the student side only (as opposed to both sides in (Laine and Aila, 2016)). For Mean Teacher we use $\alpha = 0.97$ decay rate in the Exponential Moving Average (EMA) of the student's weights. For all other hyper-parameters we reuse the values from Tarvainen and Valpola (2017) unless mentioned otherwise.

Like in Tarvainen and Valpola (2017), we use $\| \cdot \|^2$ for divergence in the consistency loss. Similarly, we ramp up the consistency cost $\lambda$ over the first 5 epochs from 0 up to it's maximum value of 100 as

done in Tarvainen and Valpola (2017). We use cosine annealing learning rates with no learning rate ramp up, unlike in the original MT implementation (Tarvainen and Valpola, 2017). Note that this is similar to the same hyperparameter settings as in Tarvainen and Valpola (2017) for ResNet[2]. We note that we use the exact same hyperparameters for the Π and MT models in each experiment setting. In contrast to the original implementation in Tarvainen and Valpola (2017) of CNN experiments, we use SGD instead of Adam.

**Understanding Experiments in Sections 3.2, 3.3**   We use the 13-layer CNN with the short learning rate schedule. We use a total batch size of 100 for CNN experiments with a labeled batch size of 50 for the Π and Mean Teacher models. We use the maximum learning rate $\eta_0 = 0.1$. For Section 3.2 we run SGD only for 180 epochs, so 0 learning rate cycles are done. For Section 3.3 we additionally run 5 learning rate cycles and sample pairs of SGD iterates from epochs 180-330 corresponding to these cycles.

**CIFAR-10 CNN Experiments**   We use a total batch size of 100 for CNN experiments with a labeled batch size of 50. We use the maximum learning rate $\eta_0 = 0.1$.

**CIFAR-10 ResNet + Shake-Shake**   We use a total batch size of 128 for ResNet experiments with a labeled batch size of 31. We use the maximum learning rate $\eta_0 = 0.05$ for CIFAR-10. This applies for both the short and long schedules.

**CIFAR-100 CNN Experiments**   We use a total batch size of 128 with a labeled batch size of 31 for $10k$ and $50k$ label settings. For the settings $50k+500k$ and $50k+237k^*$, we use a labeled batch size of 64. We also limit the number of unlabeled images used in each epoch to $100k$ images. We use the maximum learning rate $\eta_0 = 0.1$.

**CIFAR-100 ResNet + Shake-Shake**   We use a total batch size of 128 for ResNet experiments with a labeled batch size of 31 in all label settings. For the settings $50k+500k$ and $50k+237k^*$, we also limit the number of unlabeled images used in each epoch to $100k$ images. We use the maximum learning rate $\eta_0 = 0.1$. This applies for both the short and long schedules.

## A.10   DOMAIN ADAPTATION

We apply fast-SWA to the best experiment setting MT+CT+TFA for CIFAR-10 to STL according to French et al. (2018). This setting involves using confidence thresholding (CT) and also an augmentation scheme with translation, flipping, and affine transformation (TFA).

We modify the optimizer to use SGD instead of Adam (Kingma and Ba, 2015) and use cosine annealing schedule with $\ell_0 = 600, \ell = 550, c = 50$. We experimented with two fast-SWA methods: averaging weights once per epoch and averaging once every iteration, which is much more frequent that averaging every epoch as in the semi-supervised case. Interestingly, we found that for this task averaging the weights in the end of every iteration in fast-SWA converges significantly faster than averaging once per epoch and results in better performance. We report the results in Table 7.

We observe that averaging every iteration converges much faster (600 epochs instead of 3000) and results in better test accuracy. In our experiments with semi-supervised learning averaging more often than once per epoch didn't improve convergence or final results. We hypothesize that the improvement from more frequent averaging is a result of specific geometry of the loss surfaces and training trajectories in domain adaptation. We leave further analysis of applying fast-SWA to domain adaptation for future work.

**Implementation Details**   We use the public code[3] of French et al. (2018) to train the model and apply fast-SWA. While the original implementation uses Adam (Kingma and Ba, 2015), we use stochastic gradient descent with Nesterov momentum and cosine annealing learning rate with $\ell_0 = 600, \ell = 550, c = 100$ and $k = 100$. We use the maximum learning rate $\eta_0 = 0.1$ and momentum

---

[2]We use the public Pytorch code https://github.com/CuriousAI/mean-teacher as our base model for the MT model and modified it for the Π model.

[3]https://github.com/Britefury/self-ensemble-visual-domain-adapt.git

Table 7: Domain Adaptation from CIFAR-10 to STL. VADA results are from (Shu et al., 2018) and the original SE$^*$ is from French et al. (2018). SE is the score with our implementation without fast-SWA. fast-SWA $^1$ performs averaging every epoch and the final result is obtained at epoch 3000. fast-SWA $^2$ performs the averaging every *iteration* and the final result is obtained at epoch 600.

| Method | VADA | SE$^*$ | SE | SE + fast-SWA $^1$ | SE + fast-SWA $^2$ |
|---|---|---|---|---|---|
| Test Error | 20.0 | 19.9 | 18.1 | 17.1 | **16.8** |

0.9 with weight decay of scale $2 \times 10^{-4}$. We use the data augmentation setting MT+CF+TFA in Table 1 of French et al. (2018) and apply fast-SWA. The result reported is from epoch 4000.

