# OpenReview forum: "There Are Many Consistent Explanations of Unlabeled Data: Why You Should Average"
_ICLR.cc/2019/Conference_

### Official Review · AnonReviewer3 · 2018-11-02
**Nice read + questions**

**Rating:** 6
**Confidence:** 1

**Review:**

The paper is nice thread, easy to follow.

The paper proposed to apply SWA (Stochastic Weight Averaging) Izmailov et al. 2018 to the semi-supervised approached based on consistency regularization. The paper first describes the related work nicely and offers a succinct explanation of two semi-supervised approaches they study. The paper then present an analysis on SGD trajectories of these 2 approaches, drawing comparisons with the supervised training and then building a case of why SWA is a valid idea to apply. The analysis section is very well described, the theoretical explanations are easy to follow and Figure 1, Figure 2 are really helpful to understand this analysis.

Overall, the paper offers a useful insight into semi-supervised model trainings and offers recipe of converging to supervised results which is a valid contribution.

I have following questions to the authors:
1. Did the authors do the analysis and apply SWA on ImageNet training besides Cifar-10 and Cifar-100
2. The accuracy number reported in abstract (5.0% error) is top-1 error or top-5 error? I think it's top-5 but explicit mention would be great.
3. In section 3.2, authors offer an analysis by chosing epoch 170, 180. How are these epochs chosen?
4. In section 3.1, authors consider a simple model version where only small additive perturbations to student inputs are applied. Is this a practical setup i.e. is this ever the case in actual model training?
5. In section 3.3, pg 6, do authors have intuition into why weight averaging has better improvement (1.18) vs ensembling (0.94)?
6. In section 5.2, page 8 , can authors provide their intuition behind the results: "We found that the improvement on VAT is not drastic – our base implementation obtains 11.26% error where fast-SWA reduces it to 10.97%" - why did fast-SWA not improve much?

---

> ### Author Response · Authors · 2018-11-19
> **Response to Reviewer 3**
>
> Thank you for your support and thoughtful questions. Below we address your questions:
>
> 1. We have not yet been able to reproduce the baseline (without SWA) results for the MT model. We have been in touch with the authors of [1] to replicate these results. ImageNet experiments have also been difficult to run due to limited computational resources. Semi-supervised learning is more computationally intense than standard supervised training, which amplifies the computational difficulties and expense in running ImageNet experiments.
>
> 2. All the errors reported in the paper are top-1 errors. We will make this more explicit in the updated draft.
>
> 3. The epochs 170-180 are the last 10 epochs of training. We select these epochs as we are interested in the regime when the training has converged to the neighbourhood of the optimum, rather than the behavior during early iterations. We argue that in this regime SGD explores the set of possible solutions instead of converging to a single solution.
>
> 4. The typical setup uses perturbations from the data augmentations (random translation and flipping) and from dropout. The space of images is highly structured, and as such we believe that the more targeted perturbations of translation and flipping are more efficient at enforcing meaningful consistencies between teacher and student. In the 3072-dimensional input space, random perturbations will have a low projection onto these more meaningful directions.
>
> 5. The difference between the results of ensembling and averaging weights is sufficiently minor that the ordering could be different had we used a different dataset or architecture. We focus on weight averaging since ensembling N models results in N-fold increase in the number of computations at test time. Note that Izmailov et al. (Averaging Weights Leads to Wider Optima and Better Generalization, 2018, section 3.5) provides an argument that weight averaging approximates ensembling given that the averaged models are close in the weight space.
>
> 6. In Figure 2b we found that different methods can achieve different gains from averaging. Our empirical analysis in section 3 is focused on the Pi model and Mean Teacher, as the training trajectories for VAT are different, due to adversarial perturbations. Since random perturbations in Pi and MT lead to heterogeneous solution spaces, the gains from averaging could be greater in these model classes due to capturing a greater diversity of models.
>
> [1] Tarvainen and Valpola. Mean teachers are better role models: Weight-averaged consistency targets improve semi-supervised deep learning results. NIPS, 2017

---

### Official Review · AnonReviewer2 · 2018-11-03
**Very thorough analysis but limited novel contribution**

**Rating:** 8
**Confidence:** 4

**Review:**

OVERVIEW:
The paper looks at the problem of self-supervised learning using consistency-enforcing approaches. Their main contributions are two-fold:
1. Analysis to understand current state-of-the-art methods for self-supervised learning, namely the Mean Teacher model (MT) by Tarvainen and Valpola (2017) and the \Pi model (Laine and Aila, 2017). They show a theoretical analysis (Sec.3.1) of a simplified version of the \Pi model and show that it reaches flatter minima leading to good generalization. They show an analysis of the SDG trajectories (Sec. 3.2) that shows how these self-supervised models achieve flatter and lower minima compared to a fully supervised approach. They also provide an intuitive explanation to explore more solutions along the SGD trajectory. Finally, in Sec.3.3, they also discuss how ensembling and weight averaging help get better solutions.
2. Fast-SWA, which is a tweak to the SWA procedure (Izmailov et al, 2018) that averages models in the weight space along the SGD trajectory with a cyclical learning rate.
They show good performance on CIFAR-10 and CIFAR-100 with their proposed Fast-SWA.

PROS:
1. The paper contains a lot of empirical analysis explaining the behavior of these models and providing intuition about the optimization leading to their proposed solution. The problem and experiments are very organized and explained very well.
2. Exhaustive experiments, plots and tables showing very good performance on the standardized benchmark.

CONS:
1. The novel contribution (as I see it) is in the theoretical analysis of Sec. 3.1 & A.5 and the Fast-SWA procedure. The Fast-SWA is a minor tweak to the regular SWA. The theoretical analysis is the main novelty and it is hidden away in the appendix ! Also, the results seems to be derived on the basis of Avron and Toledo and the authors' contribution relative to that is not clear. Also, what is the difference between the regular \Pi model and simplified \Pi model and how big a difference does this make in your theory ?
2. Can the Fast SWA be used directly say while supervised training of ImageNet ? Or is it applicable only to self-supervised problems ? Comments on the generalizability of this contribution might help increase novelty.

OVERALL:
I like the thorough analysis and good results of the paper. The novelty being a little weak results in the final rating of 7.5 (rounded up to 8, subject to change depending on other reviewers).

---

> ### Author Response · Authors · 2018-11-19
> **Response to Reviewer 2**
>
> We appreciate your supportive and thoughtful review. Our responses are below:
>
> 1. Novelty can have many forms, and in this paper the main novelty, which we believe to be very significant, and also rare, is a thorough conceptual exploration of how loss geometry interacts with training procedures, particularly for semi-supervised learning, leading to several meaningful insights. Since many settings of neural network weights lead to essentially no loss, it is of foundational importance to understand how the geometric properties of a solution affect generalization. While conceptual papers can be more difficult to assess, they are often highly impactful, such as the ICLR paper by Zhang et. al (2016) on rethinking generalization [1].
>
> In addition to these conceptual advances, we do also propose simple algorithms, combined with very strong results on many thorough experiments. In this context, we view simplicity as a strength. There is sometimes a temptation to propose complicated approaches that can appear highly novel, but are not adopted because similar results can be achieved by simpler alternatives.  It is our contention that the extensive strong results in our paper combined with a simple algorithm, and a novel conceptual understanding (which is rare), are a real service to the community.
>
> As you note we also make novel theoretical contributions, some of which are in the appendix. We will highlight some of this material more in the main text. In addition to the novel theoretical and methodological contributions, there is also a novel empirical analyses in sections 3.2-3.3.
>
>
> In the simplified \Pi model, we consider small additive input perturbations to the inputs whereas in the full \Pi model we use random translations and horizontal flips of the inputs, and dropout perturbations on the weights. Tarvainen & Valpola (2017) showed that dropout could be removed without much degradation in performance. We view the random translations to be more targeted perturbations that lie along directions of the image manifold. This case is referred to in a footnote of the appendix section A.5. We mentioned the main results from the theoretical analysis in the main text but keep the proof details in the appendix due to space limitation. We will bring forward key parts to the main text for clarity.
>
> 2. Yes, (fast-)SWA can be used on purely supervised problems. SWA was used for supervised problems on CIFAR-10 and ImageNet (Izmailov, 2018) and achieved improved performance over SGD. Our paper, however, shows that the gains from weight averaging in consistency-based models are much larger than in semi-supervised learning than in supervised learning due to the geometric properties of the training trajectories and solutions discussed in Section 3.
>
> We really appreciate your strong support, and we hope that you can consider our comments on overall novelty -- across methods, experiments, theory, and conceptual understanding -- combined with strong results, in your final assessment. We are happy to answer any further questions.
>
> [1]: Zhang et. al. Understanding deep learning requires rethinking generalization. ICLR 2017.

---

### Official Review · AnonReviewer1 · 2018-11-04

**Rating:** 6
**Confidence:** 4

**Review:**

This paper proposes to apply Stochastic Weight Averaging to the semi-supervised learning context. It makes an interesting argument that the semi-supervised MT/Pi models are especially amenable to SWA since they are empirically observed to traverse a large flat region of the weight space during the later stages of training. To speed up training, the authors propose fast-SWA.

Secition 3.2 is a little confusing.
- If a random direction is, with high probability, not penalized, then why is it so flat along a random direction? Or is this simply an argument for why it is not guaranteed to be penalized, and therefore adversarial rays exist? I think the claim needs to be more precise (though it remains unclear how accurate the claim would be).
- I also think that there is maybe something special about measuring the SGD-SGD ray at epochs 170/180. It coincides with the regime of training where the signal is dominated by the consistency loss. Is it possible this somehow induces a near-linear path in the parameter space? I would be interested in seeing projections of other epoch’s SGD-SGD (e.g. 170/17x) vectors onto the 170/180 SGD-SGD ray and the extend to which they are co-linear.
- It is also striking that traversing the SGD-SGD ray causes an error rate so similar to the adversarial ray for the supervised model; can the authors explain this phenomenon?
- All this being said, I find the diversity argument compelling---though what would happen if we train the model even longer? Does it keep exploring?
- Overall, I am not sure how comfortable we should be with interpreting the SGD-SGD ray results. It is important that the authors provide a convincing argument for the interpretability of the SGD-SGD ray results, as this appears to be the key to the “large flat region” claim.

I think Mandt’s paper should be cited in-text, since this is what motivates Figure 2d.

Is the benefit of Fast-SWA’s fast convergence (to a competitive/better solution than SWA) unique to semi-supervised learning? Or can it be demonstrated by fully-supervised learning too? Given the focus on the semi-supervised regime, I would prefer if what the authors are proposing is, in some sense, special to the semi-supervised regime.

Table 1 is confusing to read. I just want to see a comparison between with and without using fast-SWA, *with all else kept equal*. Is the intention to compare “Previous Best CNN” and “Ours CNN”? Is this a fair comparison?

Pros:
+ Interesting story
+ Good empirical performance
Cons:
- Unclear whether the story is entirely correct

If the authors can provide a convincing case for the interpretability of the SGD-SGD results, I am happy to raise my score.

---

> ### Author Response · Authors · 2018-11-19
> **Response to Reviewer 1**
>
> Thank you for your thoughtful and supportive feedback. We address your questions below.
>
> 1. Regarding the argument in section 3.2:
>
> In section 3.2 we discuss the behavior of train and test error along different types of rays: SGD-SGD, random, and adversarial rays. We analyze the error along SGD-SGD rays for two reasons. Firstly, in fast-SWA we are averaging solutions traversed by SGD so the rays connecting SGD iterates serve as a proxy for the space we average over. Secondly, we are interested in evaluating the width of the solutions that we explore during training which we expect will be improved by the consistency training as discussed in section 3.1 and A.6. We do not expect width along random rays to be very meaningful because there are many directions in the parameter space that do not change the network outputs (see e.g. [2, 4, 5]). However, by evaluating SGD-SGD rays, we can expect that these directions corresponds to meaningful changes to our model because individual SGD updates correspond to directions that change the predictions on the training set. Furthermore, we observe that different SGD iterates produce significantly different predictions on the test data.
>
> In section 3.2 we observe that along SGD-SGD directions the Pi and MT solutions are much wider than supervised solutions. On the other hand, we observe that along random and adversarial directions the difference in flatness is less pronounced. Neural networks in general are known to be resilient to noise, explaining why both MT / Pi and Supervised models are flat along random directions [1]. At the same time neural networks are susceptible to targeted perturbations (such as adversarial attacks). We hypothesize that we do not observe improved flatness for semi-supervised methods along adversarial rays because we do not choose our input or weight perturbations adversarially, but rather they are sampled from a predefined set of transformations.
>
>
> 2. Regarding the choice of epochs 170, 180 for SGD-SGD ray analysis:
> We consider epochs 170 and 180 (the last 10 epochs of training) for the SGD-SGD rays, as we are interested in the regime when the training has converged to the neighbourhood of the optimum, rather than the behavior during early iterations. We argue that in this regime SGD explores the set of possible solutions instead of converging to a single solution.
>
> Based on your suggestion, we computed the cosine similarity between the SGD-SGD rays for epoch pairs 170&175 and 175&180 using the Pi model. We measured a value of -0.065, which corresponds to an angle of 93 degrees. Thus, the path traversed by SGD late in training is rather far from linear as the weight updates between epochs 170 and 175 and between epochs 175 and 180 are almost orthogonal.
>
>
> 3. Regarding the similarity between SGD-SGD rays and adversarial rays for supervised training:
>
> SGD-SGD directions and adversarial rays are indeed related. The adversarial ray for train loss at the given point is aligned with the gradient of the train loss at this point. The directions between SGD solutions from different epochs are also obtained by combining multiple gradient steps. In particular, if we use the full dataset as our mini-batch, the ray connecting SGD solutions at epochs 170 and 171 would be aligned with the adversarial ray computed at the SGD solution for epoch 170 (but pointing in the opposite direction).
>
> Since the adversarial ray is constructed using only the derivative of the train loss at a given point -- this local derivative information says that if we perturb the weights with an infinitesimal step, the error goes up the fastest along this adversarial direction -- it is not guaranteed that along the adversarial ray, for any given distance, the error would be as large as possible. In Figure 2 (d) we observe that locally the train and test error go up more sharply along adversarial rays, but for larger distances SGD-SGD rays exhibit similar behavior.
>
> 4. Regarding training longer:
> Yes, the model continues to explore even if we train longer, if we don’t anneal the learning rate to zero. In particular note that for the results in section 3.3 we extend the training time. We run training for a total of 330 epochs using a cyclical learning rate schedule (see section A9 for the details). Further, note that in combination with SWA or fast-SWA running longer consistently leads to improved performance. For example on CIFAR-10 with 4k labeled data using MT+fast-SWA we get 10.7% test error after 180 epochs, 10.34% after 240 epochs, 9.86% after 480 epochs, and 9.05% after 1200 epochs (see Tables 2-5 in the appendix for detailed results). The fact that running fast-SWA longer improves the results suggests that SGD continues to explore diverse solutions and is demonstrated by the diversity plots in figures 2 and 7.

---

> > ### Author Response · Authors · 2018-11-19
> > **Response to Reviewer 1 (continued)**
> >
> > 5. Regarding the SGD-SGD ray analysis:
> >
> > See 1.
> >
> > 6. Regarding Mandt’s paper:
> > Thank you for the suggestion, we will include an argument explaining the behavior in Figure 2d based on Mandt’s paper [3].
> >
> > 7. Regarding fast-SWA for supervised learning:
> > In the paper we show that the exploration done by SGD late in training in semi-supervised learning is more aggressive than in supervised learning, and leading to greater benefits from averaging. Fast-SWA, which averages weights more frequently than SWA, is designed to make use of this exploration. We also obtained preliminary results suggesting that fast-SWA can significantly improve performance in a domain adaptation model that uses the consistency term (see Section 5.5). We leave a thorough analysis of fast-SWA in supervised learning and other applications, such as domain adaptation, for future work.
> >
> >
> > 8. Regarding Table 1:
> > Table 1 summarizes the results of our approach and the best previous results reported in the literature across different settings. “Previous Best CNN” and “Ours CNN” show the results of our proposed method and the best previously reported result for the 13-layer CNN architecture, which is commonly used in the literature (see section A8 for the architecture description). “Previous Best” and “Ours” show the results for the ResNet architectures, which are the best results reported in the literature overall. In both cases the comparisons are fair, as the methods are using the same architecture. Note that we also present a direct comparison between our approach and the alternatives *with everything else kept equal* in the Figure 4 and Tables 2-5.
> >
> >
> > [1]: Arora et. al, Stronger generalization bounds for deep nets via a compression approach, 2018.
> > [2]: Anonymous, Gradient Descent Happens in a Tiny Subspace, 2019.
> > [3]: Mandt et al., Stochastic Gradient Descent as Approximate Bayesian Inference, 2017.
> > [4]: Dinh et al., Sharp Minima Can Generalize for Deep Nets
> > [5]: Sagun et al., Empirical Analysis of the Hessian of Over-Parametrized Neural Networks, 2017.

---

> ### Public Comment · (anonymous) · 2018-12-18
> **Holdout size**
>
> Hi,
> I noticed that your holdout set size is 5000. This means that in practice you are using 9000 labeled examples rather than the reported 4000. As shown recently in Oliver et al (https://arxiv.org/pdf/1804.09170.pdf) in a --proper-- evaluation where only 4000 labeled examples are used, the accuracy of SSL algorithms drops considerably.
> Thus, it is not clear why you report results for the 9K case as those obtained using 4K labeled examples.
> Can you please comment, and also say what your results are when using only 4K examples (as in Oliver et al evaluation scheme).
> Thanks

---

> > ### Author Response · Authors · 2018-12-18
> > **Clarification**
> >
> > Hi, thank you for your comment. In our paper we exactly replicate the experimental setup of [1], which uses 5000 validation images, in order to directly compare our approach with the most relevant existing literature. We note [2] uses a larger validation set of 10000 images, and [3] does not discuss validation. We are aware of the work [4], but note that their paper, with respect to validation data, only argues that small validation sets make model comparison difficult (section 4.6). In their experiments (everywhere except for section 4.6) they still use a full validation set of 5000 images, and thus replicating the setup in [4] would not change our results. Moreover, code for the evaluation framework [4] was released after the submission deadline for ICLR. Additionally, in our experiments we reuse the hyper-parameters of the Mean Teacher method [1] and only tune the learning rate schedule for fast-SWA on the validation set. And in section A.3 we demonstrate that the performance is not sensitive to the choice of this learning rate schedule.
> >
> > [1] Mean teachers are better role models: Weight-averaged consistency targets improve semi-supervised deep learning results; Antti Tarvainen, Harri Valpola
> > [2] Virtual adversarial training: a regularization method for supervised and semi-supervised learning; Miyato, Takeru, Maeda, Shin-ichi, Koyama, Masanori, and Ishii, Shin
> > [3] Temporal ensembling for semi-supervised learning; Laine, Samuli and Aila, Timo
> > [4] Realistic Evaluation of Deep Semi-Supervised Learning Algorithms; Avital Oliver, Augustus Odena, Colin Raffel, Ekin D. Cubuk, Ian J. Goodfellow

---

### Public Comment · (anonymous) · 2018-10-11
**About "the best result reported in the literature"**

“... improving the best result reported in the literature (Tarvainen and Valpola, 2017) by 1.3%.” --- appeared in the introduction.

FYI, the statement above seems like outdated because the results reported in (Tarvainen and Valpola, 2017) have been surpassed by (Wei et al., 2018) for the same underlying network architecture. It is unclear how well the WGAN+consistency method of (Wei et al., 2018) could work for the Shake-Shake architecture.

Wei, X., Gong, B., Liu, Z., Lu, W. and Wang, L., 2018. Improving the Improved Training of Wasserstein GANs: A Consistency Term and Its Dual Effect. arXiv preprint arXiv:1803.01541.

---

> ### Author Response · Authors · 2018-10-12
> **Clarification**
>
> Hello,
> Thank you for your comment. In the introduction we mention the improvement over the best overall result on CIFAR-10 with 4k unlabeled data points, which is achieved using ResNet with Shake-Shake regularization and which belonged to [3]. We improve their result from 93.7% accuracy to 95% accuracy. Note that in the experiments section we also provide results for the 13-layer CNN used by [2] (the paper you mentioned). For that architecture, the best results previously reported in the literature were to the best of our knowledge achieved by [1] (90.8% as opposed to 90% for the paper [2] you mentioned). We also further improve the results from [1] on that architecture.
>
> [1] Adversarial Dropout for Supervised and Semi-Supervised Learning. Sungrae Park, Jun-Keon Park, Su-Jin Shin, Il-Chul MoonSungrae Park, Jun-Keon Park, Su-Jin Shin, Il-Chul Moon
> [2] Improving the Improved Training of Wasserstein GANs: A Consistency Term and Its Dual Effect. Wei, X., Gong, B., Liu, Z., Lu, W. and Wang, L.
> [3] Mean teachers are better role models: Weight-averaged consistency targets improve semi-supervised deep learning results. Antti Tarvainen, Harri Valpola

---

### Public Comment · ~Olivier_Grisel1 · 2018-11-10
**Consistency regularization vs noise cushioning**

Very interesting empirical study and nice results. I have two remarks / questions.

- It seems that the consistency regularization loss is directly optimizing the interlayer noise cushioning terms from the generalization bound given in:

Stronger Generalization Bounds for Deep Nets via a Compression Approach
Sanjeev Arora, Rong Ge, Behnam Neyshabur, Yi Zhang ;
Proceedings of the 35th International Conference on Machine Learning, PMLR 80:254-263, 2018.

http://proceedings.mlr.press/v80/arora18b.html

Maybe you should discuss the relation to this theoretical work in your manuscript.


- Have you tried to apply this to non-image classification problems? In particular the combination of stochastic regularization + weight averaging seemed to be important to get SOTA performance on recurrent language models:

https://github.com/salesforce/awd-lstm-lm

I am wondering if fast-SWA with its consistency loss term could improve upon the Averaged SGD + stochastic regularization combination.

Arguably, auto-regressive language modeling cannot benefit from the semi-supervised setting as it's already a self-supervised task.

---

> ### Author Response · Authors · 2018-11-22
> **Response**
>
> Dear Olivier,
>
> Thank you for your thoughtful comments.
>
> The consistency loss encourages noise stability, which is related to compressibility in [1]. In section 3.1 we argue that the consistency loss penalizes Jacobian norm of the network on the unlabeled data. We believe that the interlayer cushion is a related but different measure of noise stability. We agree that [1] is related to the discussion in section 3.1, and we will cite it in an updated version of the paper. We note, however, that consistency regularization provides little improvement when applied to labeled training data alone -- the less constrained unlabelled data is crucial to the performance of the method. Also, targeted perturbations like data augmentation have a much larger impact on performance than more isotropic perturbations like Gaussian noise and dropout. We believe that understanding these behaviours better theoretically in relationship to [1] warrants further investigation. Specifically, there may be mileage to be gained considering non-isotropic noise stability given that the inputs lie in a much lower dimensional space than the full space of pixels.
> We have looked at fast-SWA in other domains, such as RL, with good preliminary results. For the reasons given in the paper, however, semi-supervised learning is particularly compelling for these approaches, and vision benchmarks provide a clean way to thoroughly explore and evaluate these benefits. As you mention, auto-regressive methods are not as likely to benefit from consistency regularization because the predictions are already constrained by self-supervision. We conjecture, however, that sequence labeling tasks with extra unlabelled sequences may benefit from consistency regularization and fast-SWA, which is an interesting direction for future work.
>
>
> [1] Stronger Generalization Bounds for Deep Nets via a Compression Approach
> Sanjeev Arora, Rong Ge, Behnam Neyshabur, Yi Zhang; 2018.

---

### Public Comment · (anonymous) · 2018-12-18
**Holdout set**

Hi,
I noticed that your holdout set size is 5000. This means that in practice you are using 9000 labeled examples rather than the reported 4000. As shown recently in Oliver et al (https://arxiv.org/pdf/1804.09170.pdf) in a --proper-- evaluation where only 4000 labeled examples are used, the accuracy of SSL algorithms drops considerably.
Thus, it is not clear why you report results for the 9K case as those obtained using 4K labeled examples.
Can you please comment, and also say what your results are when using only 4K examples (as in Oliver et al evaluation scheme).
Thanks

---

> ### Author Response · Authors · 2018-12-19
> **Clarification (duplicated from below)**
>
> Hi, thank you for your comment. In our paper we exactly replicate the experimental setup of [1], which uses 5000 validation images, in order to directly compare our approach with the most relevant existing literature. We note [2] uses a larger validation set of 10000 images, and [3] does not discuss validation. We are aware of the work [4], but note that their paper, with respect to validation data, only argues that small validation sets make model comparison difficult (section 4.6). In their experiments (everywhere except for section 4.6) they still use a full validation set of 5000 images, and thus replicating the setup in [4] would not change our results. Moreover, code for the evaluation framework [4] was released after the submission deadline for ICLR. Additionally, in our experiments we reuse the hyper-parameters of the Mean Teacher method [1] and only tune the learning rate schedule for fast-SWA on the validation set. And in section A.3 we demonstrate that the performance is not sensitive to the choice of this learning rate schedule.
>
> [1] Mean teachers are better role models: Weight-averaged consistency targets improve semi-supervised deep learning results; Antti Tarvainen, Harri Valpola
> [2] Virtual adversarial training: a regularization method for supervised and semi-supervised learning; Miyato, Takeru, Maeda, Shin-ichi, Koyama, Masanori, and Ishii, Shin
> [3] Temporal ensembling for semi-supervised learning; Laine, Samuli and Aila, Timo
> [4] Realistic Evaluation of Deep Semi-Supervised Learning Algorithms; Avital Oliver, Augustus Odena, Colin Raffel, Ekin D. Cubuk, Ian J. Goodfellow

---

### Meta-Review · Area_Chair1 · 2018-12-16
**Interesting analysis and insights into SWA for semisupervised learning**

**Confidence:** 5
**Recommendation:** Accept (Poster)

**Metareview:**

All reviewers appreciate the empirical analysis and insights provided in the paper. The paper also reports impressive results on SSL. It will be a good addition to the ICLR program.